# BPTF regulates androgen receptor activity by enhancing chromatin accessibility and stabilizing the AR-FOXA1 interaction

Hee-Young Jeon [1,2], Sudeep Khadka [1,2], Majid Pornour[1,2], Hyunju Ryu[1,2], Hegang Chen[3], Arif Hussain[1,2,4], Hung-Ming Lam [5], Eva Corey [5], Htoo Zarni Oo [6], Martin Gleave[6], Xiaofang Che [7], Christopher Barbieri [8] & Jianfei Qi [1,2] ✉

BPTF, the scaffolding subunit of the nucleosome remodeling factor (NURF) complex, has been implicated in the progression of several malignancies, but its role in prostate cancer (PCa) remains unclear. Here, we demonstrate that BPTF is upregulated in castration-resistant prostate cancer (CRPC) and promotes disease progression. RNA-seq revealed that BPTF primarily enhances the expression of androgen receptor (AR) target genes. ChIP-seq showed that BPTF increases AR binding at promoters, enhancers and super-enhancers. ATAC-seq further demonstrated that BPTF increases chromatin accessibility to facilitate AR binding, in part through SMARCA1, a catalytic subunit of the NURF complex. Notably, BPTF/AR co-bound regions are highly enriched for FOXA1 motifs but only weakly enriched for AR motifs. We further show that BPTF forms a protein complex with AR and FOXA1, in which FOXA1 recruits the BPTF-AR complex to chromatin, while BPTF stabilizes the AR-FOXA1 interaction. Importantly, BPTF interacts with AR through its bromodomain, and a BPTF bromodomain inhibitor disrupts this interaction, impairs AR signaling and suppresses PCa cell growth. In summary, our findings establish BPTF as a critical regulator of AR activity by promoting chromatin accessibility and stabilizing the AR-FOXA1 complex, highlighting BPTF as a potential therapeutic target in prostate cancer.

Prostate cancer (PCa) is one of the most frequently diagnosed cancers in men worldwide, with metastatic PCa being the primary cause of mortality. Androgen deprivation therapy (ADT), which suppresses androgen receptor (AR) activity, is the first-line treatment for metastatic PCa. While ADT initially induces clinical remission, recurrence typically occurs within 2–3 years, leading to the development of castration-resistant prostate cancer (CRPC), a lethal stage of the disease[1]. The primary treatments for CRPC include taxane-based chemotherapy (docetaxel or carbazitaxel) and second-generation AR pathway inhibitors (such as enzalutamide, apalutamide, abiraterone)[2,3]. Although these therapies extend survival, they ultimately fail. The progression from castration-sensitive prostate cancer (CSPC) to CRPC and the failure of CRPC-directed therapies are primarily driven by reactivation of the AR transcriptional program

[1]Department of Biochemistry and Molecular Biology, University of Maryland, Baltimore, MD, USA. [2]Marlene and Stewart Greenebaum Comprehensive Cancer Center, Baltimore, MD, USA. [3]Department of Epidemiology and Public Health, University of Maryland, Baltimore, MD, USA. [4]Baltimore Veterans Affairs Medical Center, Baltimore, MD, USA. [5]Department of Urology, University of Washington, Seattle, WA, USA. [6]Department of Urologic Sciences, Vancouver Prostate Centre, University of British Columbia, Vancouver, BC, Canada. [7]Department of Medical Oncology, the First Hospital of China Medical University, Shenyang, China. [8]Department of Urology, Weill Cornell Medicine, New York, NY, USA. ✉e-mail: jqi@som.umaryland.edu

through mechanisms such as AR overexpression, mutation, alternative splicing, AR co-factor upregulation, and intratumoral androgen synthesis[4,5]. Thus, understanding the regulatory mechanisms of AR activation may reveal new therapeutic opportunities for PCa.

Bromodomain PHD finger transcription factor (BPTF) is the largest subunit of the nucleosome remodeling factor (NURF) complex, an ATP-dependent chromatin remodeler. The C-terminal region of BPTF has a PHD finger and a bromodomain (BRD), which recognize histone H3 lysine 4 methylation (H3K4me2/3) and histone acetylation, respectively[6–8]. SMARCA1 or SMARCA5 serves as the catalytic subunit of the NURF complex, mediating nucleosome sliding to regulate DNA accessibility[7–9]. BPTF is upregulated and plays an oncogenic role in multiple malignancies, including melanoma, glioma, neuroblastoma, acute myeloid leukemia, B-cell lymphoma, and cancers of the pancreas, colon, stomach, breast, lung, and kidney[10–22]. In addition, the *BPTF* gene fuses with *NUP98* to generate the *NUP98-BPTF* fusion, which contributes to leukemogenesis via aberrant transcriptional regulation[23]. A well-established oncogenic function of BPTF is its ability to promote c-Myc expression, activity, or chromatin binding[10–13,18–22]. However, its potential role in PCa remains unexplored.

In this study, we identify a novel role for BPTF in regulating AR signaling in PCa. Our RNA-seq, CUT&RUN ChIP-seq, and ATAC-seq analyses demonstrate that BPTF enhances chromatin accessibility and stabilizes the AR-FOXA1 complex, thereby upregulating AR target gene expression. Furthermore, the mechanisms can be disrupted using a BPTF inhibitor. Our findings establish BPTF as a promising therapeutic target for inhibiting AR signaling in PCa.

## Results

### BPTF is upregulated in CRPC and promotes PCa progression

Analysis of transcriptomic datasets from public databases revealed that *BPTF* expression levels in PCa cell lines are comparable to those in other cancer cell lines (Fig. S1A). Similarly, *BPTF* expression in primary PCa tissues is comparable to that in most other tumor types (Fig. S1B). These findings are consistent with a tumor-promoting role for BPTF across multiple cancer types[10–22]. Interestingly, analysis of three publicly available PCa datasets revealed a significant increase of *BPTF* mRNA levels in metastatic CRPC compared to primary PCa (Fig. 1A, B, Fig. S1C). We next performed immunohistochemistry staining of BPTF in PCa tissue microarrays (TMAs). BPTF staining was elevated in PCa samples with lymph node metastasis and in CRPC compared to benign prostatic hyperplasia (BPH) and primary PCa (Fig. 1C, D). Among primary PCa samples, BPTF staining was higher in high-grade tumors (Gleason score 8–10) than in low-grade tumors (Gleason score 6–7) (Fig. 1E). These findings suggest that BPTF expression is positively associated with PCa progression. To confirm the functional role of BPTF in PCa, we transduced *BPTF* shRNA (sh-1) lentivirus for 48 h to knock down BPTF expression in several PCa cell lines, including Rv1, C4-2, VCaP, and LNCaP. BPTF knockdown (KD) significantly reduced *BPTF* mRNA and protein levels (Fig. 1F, G) and markedly inhibited colony formation by these PCa cells compared to the non-targeting control shRNA (Fig. 1H, I). A separate *BPTF* shRNA (sh-2) showed similar results (Fig. S1D–G). The upregulation of BPTF in CRPC suggests that BPTF may play a role in CRPC progression. To test this possibility, we upregulated endogenous BPTF expression in Rv1 cells using the CRISPR activation (CRISPRa) Synergistic Activation Mediator (SAM) system[24]. In this approach, SAM (a potent transcriptional activation complex) was directed to the *BPTF* promoter by CRISPR-Cas9-mediated guidance. Compared to a non-targeting control sgRNA, two different sgRNAs targeting the *BPTF* promoter led to increased levels of both *BPTF* mRNA and protein (Fig. 1J, K), along with increased levels of dCas9 fusion protein and RNA polymerase II (Pol II) at the *BPTF* promoter, as revealed by ChIP-qPCR analysis (Fig. 1L). These results demonstrate that the CRISPRa approach successfully upregulated endogenous BPTF expression. BPTF upregulation slightly

increased Rv1 cell colony formation (~20%) in the presence of androgen, but increased colony formation by over 2-fold under androgen-deprived conditions (Fig. 1M, N). This suggests that BPTF overexpression confers resistance to androgen deprivation. Collectively, these findings indicate that BPTF is upregulated in CRPC and plays a role in promoting PCa cell growth.

### BPTF primarily regulates AR target gene expression in PCa

To investigate the mechanisms how BPTF promotes PCa cell proliferation, we transduced Rv1 cells with *BPTF* shRNA (sh-1) lentivirus for 48 h, followed by RNA-seq analysis. Efficient KD of BPTF was confirmed at both the mRNA (Fig. 2A) and protein (Fig. S2A) levels. BPTF KD resulted in the downregulation of 529 genes (Fig. 2B, blue, Supplementary Data 1) and the upregulation of 196 genes (Fig. 2B, red, Supplementary Data 1). Next, we performed bioinformatic analyses on BPTF-activated genes (i.e., genes downregulated following BPTF knockdown). Binding Analysis for Regulation of Transcription (BART) identified AR as the top-ranked transcription factor associated with these genes (Fig. 2C). Gene Ontology (GO) pathway analysis revealed androgen response as the most significantly enriched pathway (Fig. 2D). Gene set enrichment analysis (GSEA) using the androgen response gene set confirmed that AR signature genes were significantly enriched in control cells compared with BPTF-KD cells (Fig. 2E). Furthermore, GSEA using the Pathway Interaction Database (PID; 196 gene sets) also identified the AR pathway as the most enriched in control cells relative to BPTF-KD cells (Fig. 2F). Consistently, heatmap of RNA-seq data showed that BPTF KD reduced the expression of AR hallmark target genes (Fig. 2G). Together, these results suggest that BPTF positively regulates AR activity in Rv1 cells. Similarly, we performed bioinformatic analyses on BPTF-repressed genes (i.e., genes upregulated following BPTF knockdown). BART analysis identified EZH2 as the top-ranked transcriptional regulator associated with these genes (Fig. S2B). GO pathway analysis revealed TNF-α signaling via NF-κB as the most significantly enriched pathway (Fig. S2C), whereas GSEA using PID gene sets did not identify any significantly enriched pathways (Fig. S2D). These results suggest that BPTF activates or represses distinct transcriptional programs.

Notably, the bioinformatic analyses consistently suggest that BPTF primarily enhances the expression of AR target genes. To determine whether BPTF promotes AR activity in PCa cells, we examined the expression levels of several AR target genes in various PCa cells after transduction with two different *BPTF* shRNA lentiviruses for 48 h. BPTF KD reduced mRNA levels of AR target genes in Rv1, C4-2, VCaP, and LNCaP cells (Fig. 2H, Fig. S2E). Moreover, BPTF KD in Rv1 and VCaP cells reduced AR target gene expression both in the presence and absence of synthetic androgen (R1881) after 24 h of treatment (Fig. 2I). Conversely, BPTF upregulation by CRISPRa increased mRNA levels of AR target genes but did not affect *AR* mRNA levels (Figs. 1J, K, 2J). BPTF KD did not significantly alter AR protein levels in various PCa cells (Fig. 2K), indicating that BPTF does not affect AR expression. Finally, we observed that *BPTF* expression positively correlates with the expression of *AR* and its target genes in human PCa samples (Fig. 2L, Fig. S2F). Together, these results support the notion that BPTF enhances AR activity in PCa.

Given the critical role of c-Myc in PCa progression[25–27] and previous studies linking BPTF to the regulation of the MYC pathway in various malignancies[10–13,18–22], we investigated whether BPTF regulates c-Myc in PCa cells. BART analysis of BPTF-activated genes (i.e., genes downregulated in BPTF-KD RNA-seq) ranked c-Myc as the 137th associated transcription factor (Fig. 2C), while the MYC pathway ranked 10th in GO analysis (Fig. 2D) and 23rd in GSEA (Fig. 2F). BPTF KD did not affect *c-Myc* mRNA or protein levels in various PCa cell lines (Fig. 2K, S2G). Moreover, mRNA levels of c-Myc target genes remained unchanged upon BPTF KD in these PCa cell lines (Fig. S2H–K). In contrast, BPTF KD in SW620 colorectal cancer cells, where BPTF has

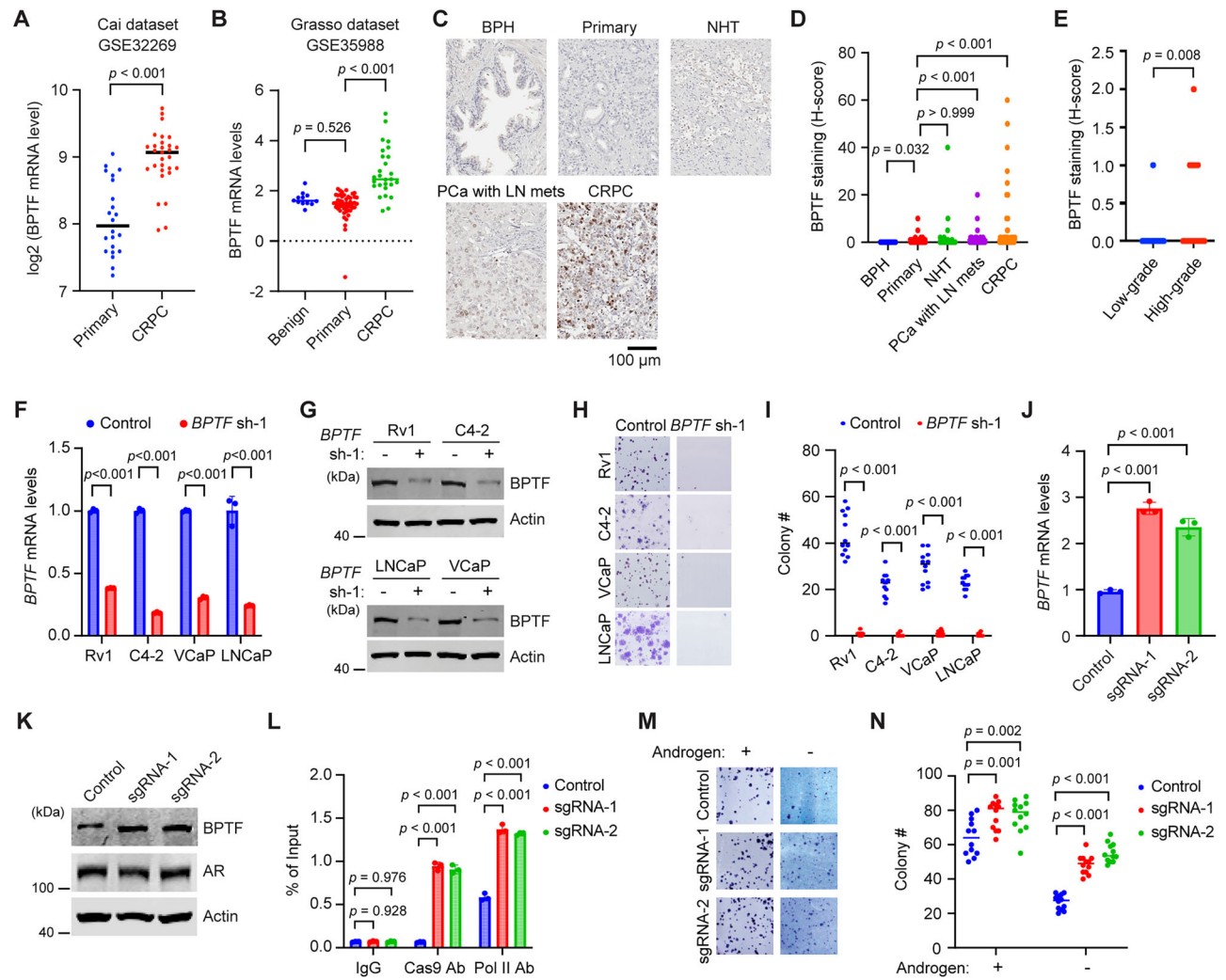

**Fig. 1 | BPTF expression is upregulated in CRPC and promotes PCa cell proliferation. A**, **B** *BPTF* mRNA expression levels in primary PCa and CRPC samples, as shown in the indicated GEO datasets. **A** primary PCa (n = 22) and CRPC (n = 29); **B** benign prostate tissue (n = 12), primary PCa (n = 49), and CRPC (n = 27). **C**, **D** Immunohistochemistry staining of BPTF in PCa tissue microarrays. **C** Representative images. BPTF, brown; nuclei, blue. **D** Quantification of BPTF staining. Sample numbers: benign prostatic hyperplasia (BPH, n = 56), primary PCa (n = 247), PCa treated with neoadjuvant hormone therapy (NHT, n = 74), PCa with lymph node metastasis (n = 45), and CRPC (n = 101). **E** BPTF staining in the low-grade (n = 64) and high-grade (n = 42) primary PCa samples defined by Gleason score (low-grade: G6-7; high-grade: G8-10). **F**, **G** BPTF KD in PCa cells transduced with Control or *BPTF* shRNA. *BPTF* mRNA (**F**) and protein (**G**) levels were assessed by qRT-PCR and western blot, respectively. **H**, **I** Colony formation assays in PCa cells following BPTF KD. Representative images (**H**) and quantification (**I**) are shown.

Colonies were counted in 12 high-power fields. **J**, **K** CRISPRa-mediated upregulation of BPTF in Rv1 cells using sgRNAs targeting the *BPTF* promoter. BPTF expression was assessed by qRT-PCR (**J**) and western blots (**K**). **L** ChIP-qPCR analysis of dCas9 fusion protein and Pol II enrichment at the *BPTF* promoter following CRISPRa in Rv1 cells. **M**, **N** Colony formation assays in Rv1 cells with CRISPRa-mediated BPTF overexpression under the presence (FBS) or absence (CS-FBS) of androgen conditions. Representative images (**M**) and quantification (**N**) are shown. Data are representative of three independent biological replicates (**F**–**N**). Data are presented as mean ± SD (**F**, **J**, **L**). Statistical significance was determined using a two-tailed unpaired Student's t-test (**A**, **F**, **I**); One-way ANOVA (two-sided) with Tukey's multiple comparison test (**B**, **J**, **L**); Kruskal–Wallis test with Dunn's multiple comparisons test (**D**); Two-tailed Mann–Whitney test (**E**); Two-way ANOVA (two-sided) with Tukey's multiple comparison test (**N**). Source data are provided as a Source data file.

been reported to regulate c-Myc activity[22], led to reduced expression of these c-Myc target genes (Fig. S2L). These findings suggest that the MYC pathway is not a primary target of BPTF in PCa cells, although we cannot exclude the possibility that BPTF may regulate a subset of c-Myc target genes, as indicated by the modest but detectable enrichment of MYC-related signatures in our analyses.

**BPTF interacts with AR**

Since BPTF KD did not affect AR levels, we hypothesized that BPTF may interact with AR and modulate its transcriptional activity. Supporting this hypothesis, our AR RIME (Rapid Immunoprecipitation Mass spectrometry of Endogenous Proteins) analysis identified BPTF as one of the proteins co-precipitated with AR in Rv1 cells, based on mass

spectrometry of the AR-associated chromatin complex (Supplementary Data 2). To validate this, we performed co-immunoprecipitation (co-IP) assays and found the co-IP of endogenous AR with BPTF in Rv1 cells, indicating an AR-BPTF interaction (Fig. 3A). To map the domains mediating this interaction, we generated seven serial truncation fragments of BPTF (F1 to F7, Fig. S3A) and co-transfected each with AR in 293 T cells. Co-IP analysis revealed that AR co-precipitated with the C-terminal fragment of BPTF (F7), which contains the PHD finger and bromodomain (BRD) regions (Fig. 3B). To determine whether the PHD finger or BRD domain is involved in the interaction with AR, we generated constructs encoding the C-terminal fragments of BPTF, either wild-type (F6 + 7) or a truncation mutant with the BRD domain deleted (F6 + 7−BRD) (Fig. S3A), and tested them for co-IP with

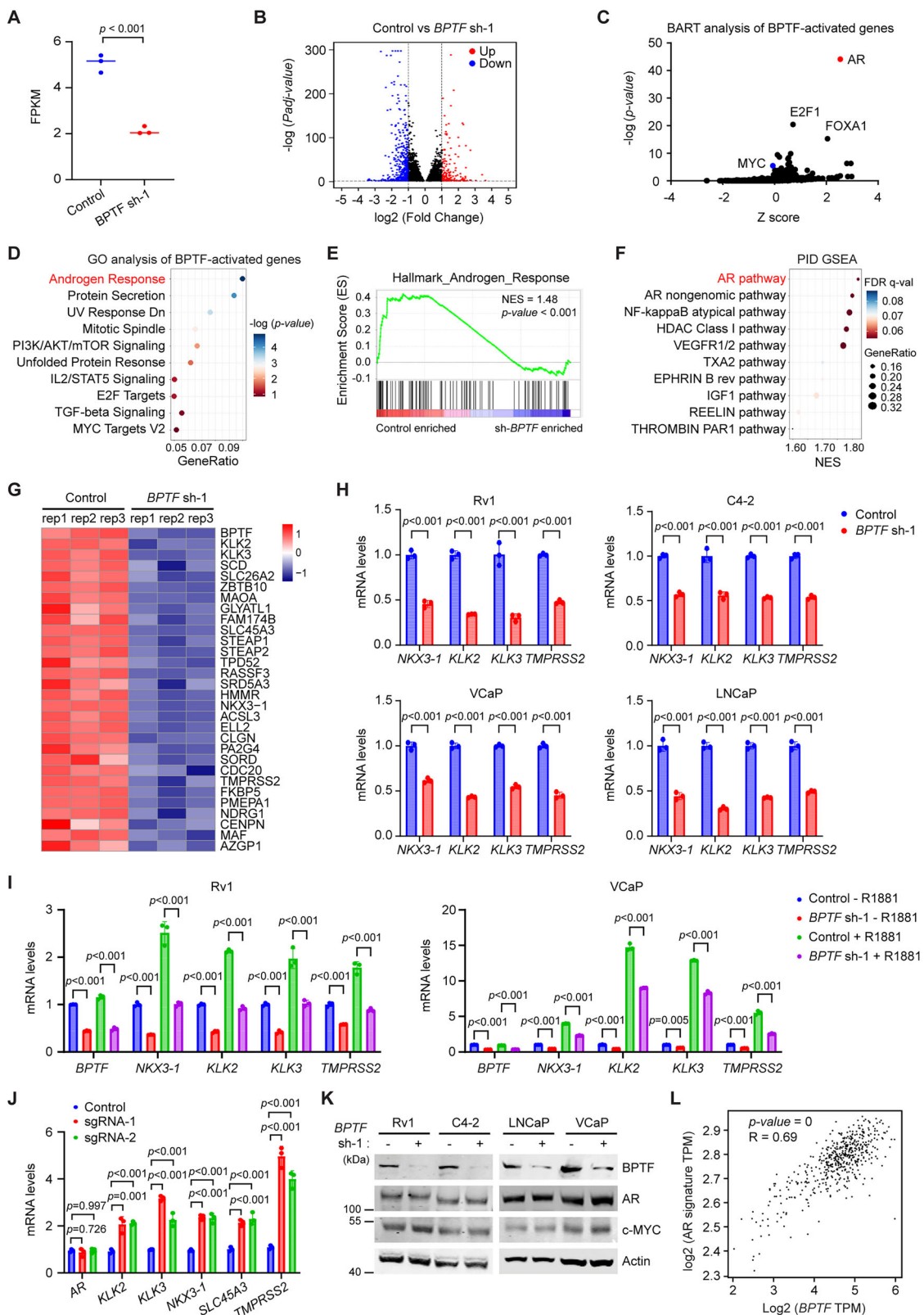

AR. The results showed that AR co-precipitated with wild-type BPTF fragment but not with the BRD-deleted BPTF fragment (Fig. 3C), indicating that the BRD is required for interaction with AR. Similarly, co-IP analysis of AR truncation mutants showed that the ligand-binding domain (LBD) of AR co-precipitated with the C-terminal BPTF fragment (Fig. 3D). Together, these results reveal that the BPTF-BRD and AR-LBD domains mediate the BPTF-AR interaction. Since AR-LBD

interacts with BPTF, we next examined whether androgen (R1881) or the AR antagonist enzalutamide (ENZ) affects the AR-BPTF interaction. C4-2 cells were cultured under androgen-deprived conditions for 3 days, followed by treatment with 1 nM R1881 for 24 h. Co-IP analysis revealed increased association of BPTF with AR upon androgen treatment (Fig. 3E). In contrast, treatment of C4-2 cells with enzalutamide (5 μM) for 24 h reduced BPTF-AR interaction (Fig. 3F). These results

**Fig. 2 | BPTF primarily upregulates AR activity in PCa cells. A** Validation of BPTF KD efficiency in RNA-seq samples of Rv1 cells. **B** Volcano plot showing the differentially expressed genes (DEGs) identified by RNA-seq of control and BPTF-KD Rv1 cells. Significantly downregulated genes were shown in blue ($\log_2$ [Fold Change] < −1, *Padj* < 0.05), and significantly upregulated genes were shown in red ($\log_2$ [Fold Change] > 1, *Padj* < 0.05). **C** BART analysis of downregulated genes ($\log_2$ [Fold Change] < −1, *Padj* < 0.05) after BPTF KD, predicting transcription factors underlying the altered gene expression. **D** GO analysis of downregulated genes ($\log_2$ [Fold Change] < −0.8, *Padj* < 0.05) after BPTF KD based on the MSigDB hallmark gene set. **E** GSEA of DEGs based on the androgen response gene set. **F** GSEA of DEGs based on the Pathway Interaction Database (PID), which includes 196 gene sets for various pathways. Only pathways enriched in control cells were shown. **G** Heatmap showing the decreased expression of AR hallmark target genes in BPTF-KD RNA-seq samples. **H** qRT-PCR analysis of representative AR target genes following BPTF KD in various PCa cells. **I** qRT-PCR analysis of AR target genes in BPTF-KD Rv1 (left panel) and VCaP (right panel) cells cultured in the presence (+R1881) or absence (−R1881) of androgen. **J** qRT-PCR analysis of AR target genes following CRISPRa-mediated upregulation of BPTF in Rv1 cells. **K** Western blots showing AR and c-Myc protein levels in BPTF-KD PCa cells. **L** Pearson correlation analysis of *BPTF* expression with AR signature gene expression in TCGA PCa dataset (n = 492). RNA-seq data were generated from 3 biological replicates (**A–G**). Data are representative of three independent biological replicates (**H–K**). Data are presented as mean ± SD (**H–J**). Statistical significance was determined using a two-tailed unpaired Student's t-test (**A, H**); Two-way ANOVA (two-sided) with Tukey's multiple comparison test (**I**); One-way ANOVA (two-sided) with Tukey's multiple comparison test (**J**). Source data are provided as a Source data file.

support the involvement of AR-LBD in mediating the interaction with BPTF.

## BPTF/AR peaks co-localize on chromatin

To evaluate the genome-wide BPTF-AR interaction in Rv1 cells, we performed CUT&RUN ChIP-seq analysis using BPTF or AR antibodies. This revealed 24,483 BPTF peaks and 19,907 AR peaks, with no significant peaks in IgG control ChIP-seq (Fig. 3G, J). The majority of BPTF and AR peaks were located at intergenic, intronic, and promoter/5′UTR regions (Fig. 3H, K, left panel). To further assess the location of BPTF and AR peaks, we used CUT&RUN ChIP-seq peaks of H3K27ac and H3K4me1 in Rv1 cells to define active enhancers (H3K27ac + / H3K4me1 + , located more than 2 kb upstream or downstream from the transcriptional start site) and active promoters (H3K27ac + , located within 2 kb upstream or downstream from the transcriptional start site). Notably, 42.7% of BPTF peaks were located at active promoters and 24.4% at active enhancers (Fig. 3H, right panel). In comparison, 36.8% of AR peaks were found at active promoters and 40% at active enhancers (Fig. 3K, right panel). Motif analysis of BPTF or AR peaks using HOMER revealed enrichment of AR signaling-related motifs, such as FOXA1, androgen response element (ARE), and AR half-site (Fig. 3I, L, Fig. S3B, C). Furthermore, we observed that 57.4 % of BPTF peaks (14,051 out of 24,483) overlapped with AR peaks, and 70.6 % of AR peaks (14,051 of 19,907) overlapped with BPTF peaks (Fig. 3M, Fig. S3D). We next performed motif and GO pathway analyses for each peak category (BPTF-only, BPTF/AR common, and AR-only). BPTF-only peaks lacked enrichment for AR signaling-related motifs (Fig. S3E). In contrast, BPTF/AR common peaks were strongly enriched for FOXA1 motifs (ranked 6th), but showed weak enrichment for AR motifs (AR half-site ranked 72nd; ARE ranked 78th) (Fig. S3F). By comparison, AR-only peaks exhibited strong enrichment for both AR and FOXA1 motifs (Fig. S3G). As expected, GO analysis of genes associated with both AR-only and BPTF/AR common peaks identified androgen response as one of the top-enriched pathways (Fig. S3I, J). Moreover, GO analysis revealed distinct pathway enrichment patterns for BPTF/AR common peaks compared to BPTF-only and AR-only peaks (Fig. S3H–J), suggesting that BPTF/AR co-occupied regions may be associated with unique biological functions. The track images showed the co-localization of BPTF and AR peaks at the promoter or enhancer regions of AR target genes such as *NKX3-1*, *SLC45A3*, *KLK2*, and *KLK3* (Fig. 3N). While the enhancers of *KLK2* and *KLK3* are well-characterized, those of *NKX3-1* and *SLC45A3* have not been extensively studied. Based on the published H3K27ac HiChIP data in Rv1 cells[28], we identified the potential enhancer regions for *NKX3-1* and *SLC45A3* that loop to their corresponding promoters (Fig. 3N). ChIP-qPCR analysis confirmed that both BPTF and AR were significantly enriched at promotor or enhancer regions of the representative AR target genes (Fig. 3O, P). Finally, ChIP-qPCR showed that CRISPRa-mediated BPTF upregulation increased the levels of both BPTF and AR at the *KLK3* enhancer and *NKX3-1* promoter (Fig. 3Q). These results collectively suggest that BPTF and AR interact and co-localize on chromatin, particularly at regulatory regions of AR target genes.

## BPTF upregulates the expression of AR target genes by promoting AR binding to promoters and enhancers

To determine whether BPTF regulates AR chromatin binding, we performed CUT&RUN ChIP-seq of AR after transduction of Rv1 cells with *BPTF* shRNA (sh-1) lentivirus for 48 h. Comparative analysis of AR peaks between control and BPTF-KD conditions revealed 2190 peaks enriched in control cells and 1213 peaks enriched in BPTF-KD cells ($p < 0.05$, Fig. 4A). Focusing on the control-enriched AR peaks, referred to hereafter as BPTF-dependent AR peaks, the heatmap showed a marked reduction in the signal of AR peaks following BPTF KD (Fig. 4B). Using H3K27ac and H3K4me1 ChIP-seq data to define active enhancers and promoters in Rv1 cells, we found that 44.9% of BPTF-dependent AR peaks were located at active enhancers, and 7.9% at active promoters (Fig. 4C). The decrease of AR peak signals following BPTF KD was more pronounced at enhancers compared to promoters (Fig. 4D). These results suggest that BPTF promotes AR binding to enhancers as well as promoters. For example, BPTF KD significantly reduced the signal of AR peaks at the enhancer or promoter regions of the AR target genes *NKX3-1*, *KLK2*, *KLK3*, and *SLC45A3* (Fig. 4E) a finding further validated by ChIP-qPCR in Rv1 and VCaP cells (Fig. 4F, Fig. S4A). Binding and Expression Target Analysis (BETA) is a software tool that integrates ChIP-seq data with differential gene expression data to infer direct target genes[29]. We used BETA to integrate the BPTF-dependent AR peaks with differentially expressed genes from BPTF-KD RNA-seq. The results showed that the BPTF-dependent AR peaks positively correlated with the downregulated genes (but not upregulated genes) after BPTF KD (Fig. 4G), suggesting these may be potential direct target genes regulated by the AR-BPTF complex. Indeed, GO analysis of these direct target genes confirmed that AR signaling was the most enriched pathway (Fig. 4H). Together, these results demonstrated that BPTF promotes the expression of AR target genes by facilitating AR binding to enhancers as well as promoters. Next, we performed similar analyses on the BPTF-repressed AR peaks (i.e., the 1213 AR peaks enriched in BPTF-KD cells). The heatmap showed increased AR binding following BPTF knockdown (Fig. S4B). Among these peaks, 43.6% were located at active enhancers and 24.5% at active promoters (Fig. S4C), accompanied by increased AR signal at both enhancers and promoters (Fig. S4D, E). BETA analysis revealed a positive correlation between BPTF-KD-enriched AR peaks and genes upregulated after BPTF KD (Fig. S4F). GO analysis of these genes identified skeletal muscle cell differentiation as the top-enriched pathway (Fig. S4G). Together, these findings suggest that BPTF directly activates or represses AR binding to distinct subsets of genes, thereby modulating different biological processes.

## BPTF upregulates expression of some AR target genes via super-enhancers

Super-enhancers (SEs) are clusters of enhancers that drive the high-level expression of genes that define cell identity. Previous studies reported that BPTF contributes to the formation of SEs that activate target gene expression in renal cell carcinoma[17]. To determine whether

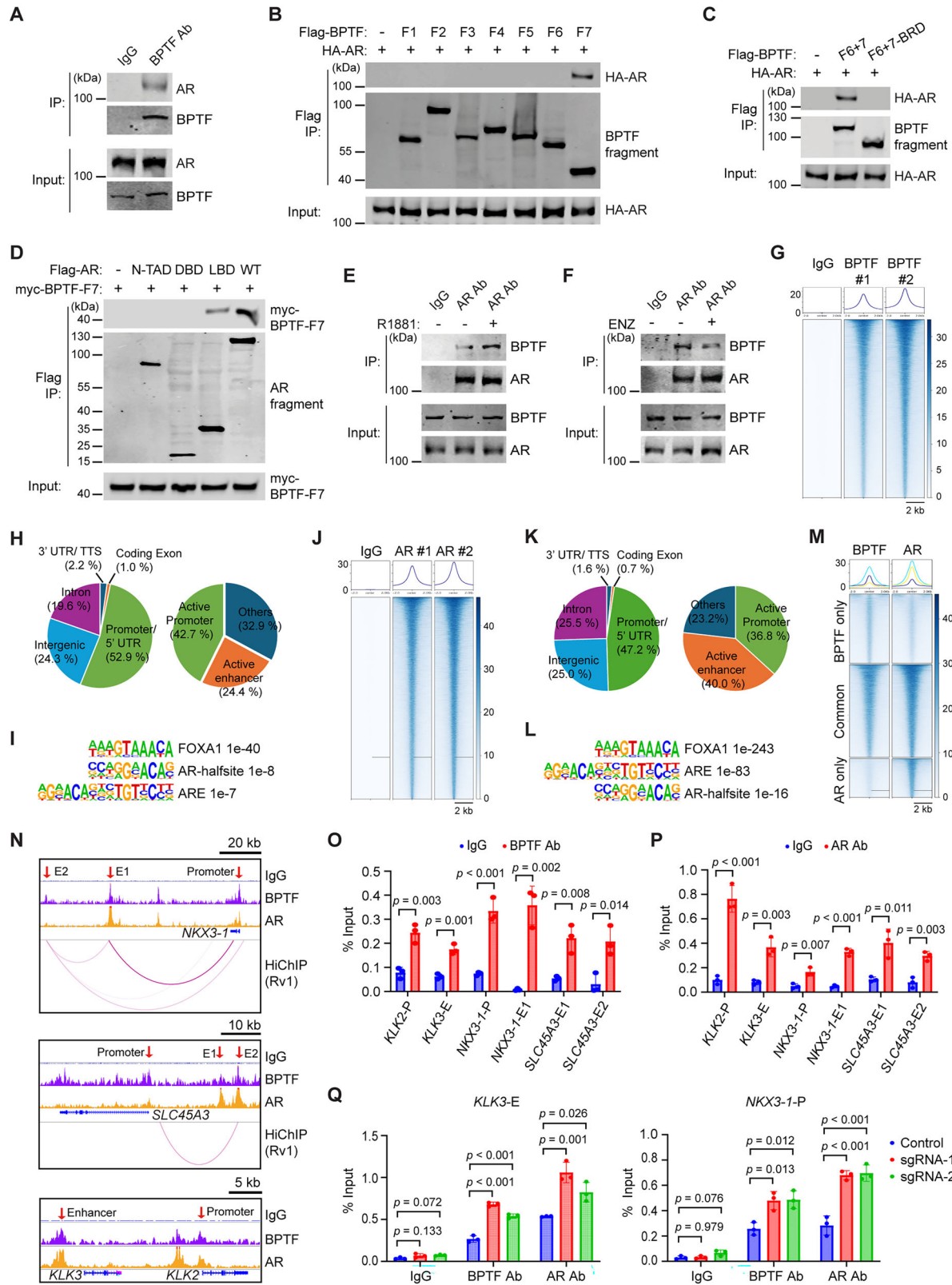

BPTF binds to SEs in Rv1 cells and influences gene expression, we applied the rank ordering of super-enhancers (ROSE) algorithm[30,31] to enhancers defined by H3K27ac and H3K4me1 ChIP-seq peaks. This analysis identified 604 SEs associated with 1001 genes (Fig. 4I). BPTF and AR peaks were highly enriched at SE regions, and BPTF KD led to a significant reduction in the signal of BPTF and AR peaks at these SE

sites (Fig. 4J, K). Among SE-associated genes (n = 1001), we identified 127 genes that met the criteria of log$_2$ FC < −0.5 or log$_2$ FC > 0.5, and padj <0.05 from BPTF-KD RNA-seq data. BPTF KD significantly reduced the expression of 80 SE-associated genes (Fig. 4L), such as *SLC45A3*, *STEAP1*, *STEAP2*, *FKBP5*, *PMEPA1*, *ELL2*, and *ZBTB10*. GO analysis showed that the 80 SE-associated genes are most strongly associated

**Fig. 3 | BPTF interacts with AR and co-localizes at AR target genes. A** Co-IP of AR with BPTF in Rv1 cells. **B** Co-IP of HA-AR with the Flag-tagged BPTF fragments in 293 T cells. **C** Co-IP of HA-AR with Flag-tagged BPTF C-terminal fragment (F6 + 7), but not with its BRD-deleted version (F6 + 7−BRD), in 293 T cells. **D** Co-IP of myc-tagged BPTF C-terminal fragment (F7) with Flag-tagged AR fragments (N-terminal transactivation domain N-TAD, DNA-binding domain DBD, Ligand-binding domain LBD, wild type WT) in 293 T cells. **E** Co-IP of BPTF with AR following R1881 treatment in C4-2 cells. **F** Co-IP of BPTF with AR following enzalutamide (ENZ) treatment in C4-2 cells. **G** Heatmap showing BPTF ChIP-seq peaks. CUT&RUN ChIP-seq was performed on Rv1 cells using a BPTF antibody or an IgG control antibody. **H** Genomic distribution of BPTF peaks. **I** HOMER motif analysis of BPTF peaks showing enrichment of FOXA1, androgen response element (ARE), and AR half-site motifs. **J** Heatmap of AR ChIP-seq peaks. CUT&RUN ChIP-seq was performed on Rv1 cells using an AR antibody or an IgG control antibody. **K** Genomic distribution of AR peaks. **L** HOMER motif analysis of AR peaks showing enrichment of FOXA1, ARE,

and AR half-site motifs. **M** Heatmap showing the overlap between BPTF peaks and AR peaks. **N** Genome browser track images showing BPTF and AR peaks at the promoter and enhancer regions of representative AR target genes. H3K27ac HiChIP data from Rv1 cells highlight potential enhancer regions for *NKX3-1* and *SLC45A3*, which form loops with their corresponding promoter region. **O, P** ChIP-qPCR analysis of BPTF (**O**) and AR (**P**) at promoter or enhancer regions of *KLK2*, *KLK3*, *NKX3−1*, and *SLC45A3* in Rv1 cells. **Q** ChIP-qPCR analysis of BPTF and AR at *KLK3* enhancer (left) and *NKX3−1* promoter (right) following CRISPRa-mediated upregulation of BPTF in Rv1 cells. CUT&RUN ChIP-seq (**G–N**) were generated from two biological replicates. All other data (**A–F**, **O–Q**) are representative of three independent biological replicates. Data are presented as mean ± SD (**O–Q**). Statistical significance was determined using a two-tailed unpaired Student's t-test (**O, P**); One-way ANOVA (two-sided) with Tukey's multiple comparison test (**Q**). Source data are provided as a Source data file.

---

with the androgen response pathway (Fig. 4M). Collectively, these findings indicate that BPTF binds to SEs to promote the expression of some AR target genes.

## BPTF regulates chromatin accessibility to promote AR binding

Since BPTF is a subunit of the NURF nucleosome remodeling complex, we hypothesized that the observed changes in AR binding following BPTF KD could be due to altered chromatin accessibility. To test this hypothesis, we performed the assay for transposase-accessible chromatin sequencing (ATAC-seq) following transduction of Rv1 cells with *BPTF* shRNA (sh-1) lentivirus for 48 h. Differential analysis of ATAC-seq peaks revealed 6,944 peaks enriched in control cells and 3,716 peaks enriched in BPTF-KD cells ($p < 0.05$; Fig. 5A). The total or differential peaks were predominantly located at intergenic, intronic, and promoter regions (Fig. 5B, Fig. S5A). BPTF KD moderately reduced global ATAC-seq signal intensity (Fig. 5C) and significantly decreased ATAC-seq signal at BPTF-dependent AR-binding sites (Fig. 5D). To identify differential transcription factor binding among the ATAC-seq peaks, we performed footprint analysis[32], which detects transcription factor binding motifs within regions protected from Tn5 transposase cleavage due to protein occupancy. Footprint analysis revealed greater enrichment of AR motifs (ranked 10th) in control cells compared with BPTF-KD cells (Fig. S5B). As expected, AR motif enrichment was further elevated (ranked 4th) at BPTF-dependent AR-binding sites in control cells relative to BPTF-KD cells (Fig. S5C). These findings suggest that BPTF promotes AR chromatin binding and protects AR motifs from Tn5-mediated cleavage, supporting the hypothesis that BPTF enhances chromatin accessibility to facilitate AR recruitment. Notably, footprint analysis also revealed increased enrichment of specific transcription factor motifs in BPTF-KD cells compared to control cells (Fig. S5B, C), suggesting that BPTF may also reduce chromatin accessibility to repress the binding of certain transcription factors. Together, these results indicate that BPTF can both enhance and repress chromatin accessibility to regulate the binding of distinct sets of transcription factors. This dual role is consistent with previous reports showing that chromatin remodeling complexes can act as both activator and repressor of gene expression, likely by modulating chromatin structure to either promote or restrict accessibility[33–36].

To further investigate the role of the NURF complex in AR target gene expression, we assessed the involvement of the catalytic subunits SMARCA1 and SMARCA5[8]. We performed RNA-seq analysis on SMARCA1-KD or SMARCA5-KD Rv1 cells after transducing *SMARCA1* or *SMARCA5* shRNA lentivirus for 48 h. GSEA of differentially expressed genes revealed enrichment of the AR pathway in control cells compared to SMARCA1-KD cells (ranked 5th, FDR = 0.662; Fig. S5D), whereas no enrichment of the AR pathway was observed in control cells relative to SMARCA5-KD cells (Fig. S5E). These findings suggest that SMARCA1 plays a more prominent role than SMARCA5 in

regulating AR activity. Consistently, RNA-seq analysis revealed that SMARCA1 KD reduced the expression of a subset of BPTF-dependent AR target genes (Fig. 5E), whereas SMARCA5 KD decreased the expression of approximately two-thirds of the same subset but increased the expression of the remaining one-third (Fig. 5F). These findings suggest that SMARCA1 and SMARCA5 have both overlapping and distinct regulatory effects on the expression of BPTF-dependent AR target genes. To test the potential redundancy between SMARCA1 and SMARCA5 on commonly regulated genes, we analyzed the expression of several AR target genes (*KLK2, KLK3, STEAP2*) following individual and combined knockdown of SMARCA1 and SMARCA5. Consistent with the RNA-seq data (Fig. 5E, F), knockdown of either SMARCA1 or SMARCA5 led to a similar reduction in the expression of these target genes (Fig. S5F). However, the combined knockdown did not result in further suppression beyond that observed with either knockdown alone (Fig. S5F). These findings do not support a redundant role for SMARCA1 and SMARCA5 in regulating these target genes. Instead, the lack of compensatory effects suggests that SMARCA1 and SMARCA5 may operate within distinct NURF complexes to regulate AR activity.

Due to the greater effect of SMARCA1 on the expression of BPTF-dependent AR target genes, we chose SMARCA1 to further evaluate its role in chromatin accessibility. ATAC-seq analysis of SMARCA1-KD Rv1 cells identified 14,928 peaks enriched in control cells and 413 peaks enriched in SMARCA1-KD cells ($p < 0.05$; Fig. 5G). SMARCA1 KD decreased the signal of ATAC-seq peaks at BPTF-dependent AR-binding sites (Fig. 5H), although the effect was less pronounced compared to BPTF KD (Fig. 5D). Further analysis identified 2394 shared ATAC-seq peaks that were decreased under both BPTF-KD and SMARCA1-KD conditions (Fig. 5I, J). Thus, 34.5% (2394/6944) of BPTF-dependent ATAC-seq peaks rely on SMARCA1. BART analysis of genes associated with these 2394 shared ATAC-seq peaks revealed AR as the top predicted transcription factor regulating these genes (Fig. 5K). To evaluate the interaction of SMARCA1 with the BPTF-AR complex on chromatin, we performed CUT&RUN ChIP-seq on Rv1 cells using a SMARCA1 antibody. SMARCA1 peaks overlapped with 47.2% of the BPTF/AR shared peaks (Fig. S5G). GO analysis showed that the androgen response pathway was the top-enriched pathway among genes associated with AR/BPTF/SMARCA1 peaks (Fig. S5H), while other enriched pathways were largely similar to those associated with AR/BPTF peaks (Fig. S3I). The motifs enriched in AR/BPTF/SMARCA1 peaks were largely similar to those in AR/BPTF peaks, except for the FOXA1 motif (Figs. S5I, S3F). Notably, while the FOXA1 motif was strongly enriched at AR/BPTF peaks (Fig. S3F), it was not among the top motifs at AR/BPTF/SMARCA1 peaks (ranked 27th; Fig. S5I), suggesting that FOXA1 may play a lesser role in recruiting the AR/BPTF/SMARCA1 complex. Together, these results indicate that SMARCA1 co-occupies chromatin with AR and BPTF and contributes to BPTF-dependent chromatin accessibility, thereby promoting AR target gene expression.

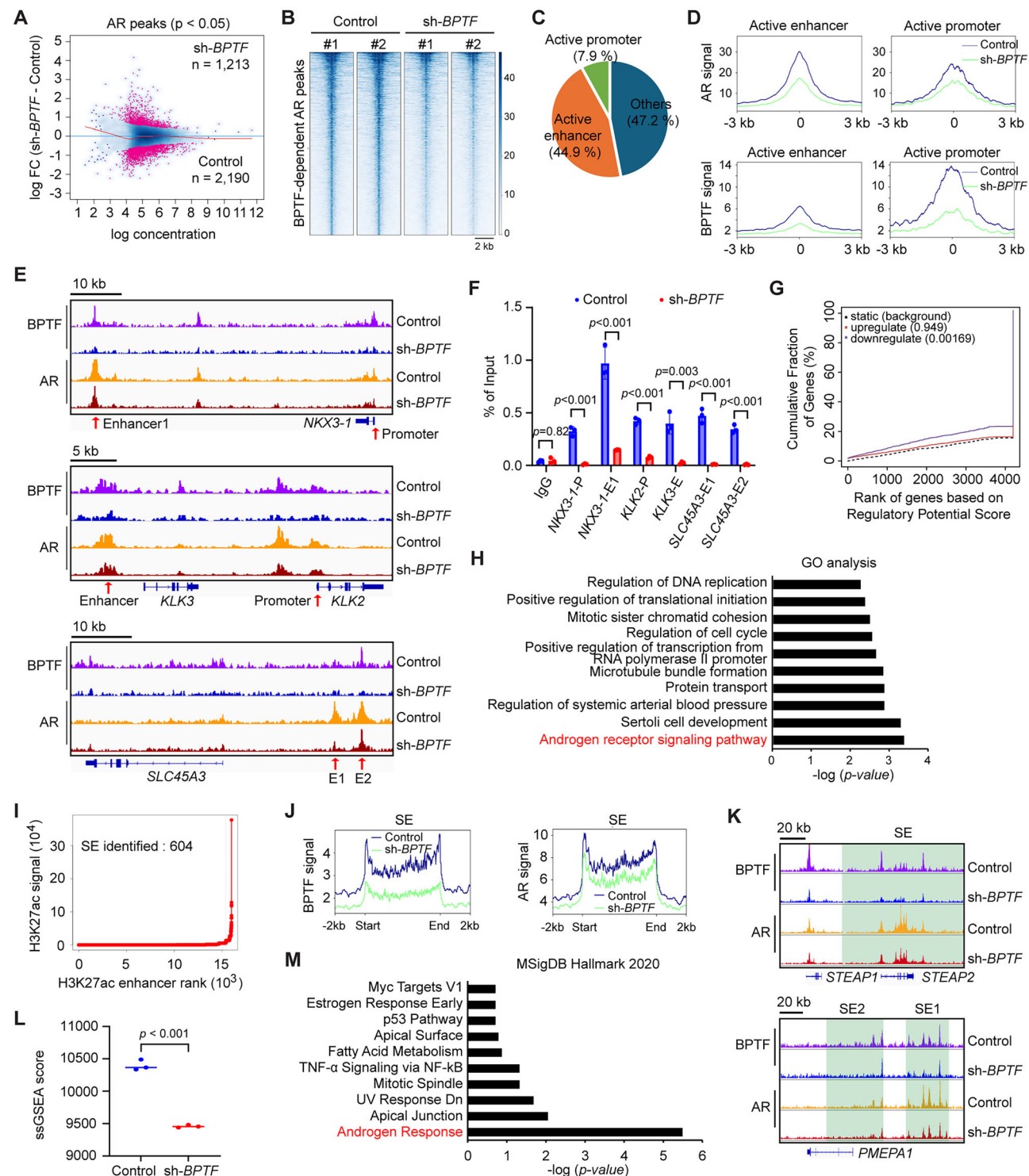

## BPTF promotes AR binding to FOXA1 on chromatin

The catalytic subunit of the NURF complex regulates expression of certain BPTF-dependent AR target genes (Fig. 5E, F), suggesting that BPTF may use an additional mechanism to regulate expression of other AR target genes. Motif analysis of BPTF/AR shared peaks revealed strong enrichment of the FOXA1 motif, but not the AR motif (Fig. S3F). Given FOXA1's role as a pioneer factor that regulates chromatin accessibility and transcriptional activity of AR[37,38], we tested whether FOXA1 plays a role in BPTF-mediated regulation of AR. We performed RNA-seq analysis on FOXA1-KD Rv1 cells 48 h after *FOXA1* shRNA

lentiviral transduction and found that FOXA1 KD reduced the expression of a subset of BPTF-dependent AR target genes (Fig. 6A). However, FOXA1 KD did not affect the mRNA or protein levels of either BPTF or AR (Fig. S6A, B). To test whether BPTF interacts with FOXA1, we performed co-IP assays and found that endogenous FOXA1 co-precipitated with BPTF in Rv1 and C4-2 cells (Fig. 6B, Fig. S6C). To determine domains mediating BPTF-FOXA1 interaction, we co-expressed FOXA1 with BPTF truncation fragments in 293 T cells and found that HA-FOXA1 co-precipitated with BPTF fragments F3, F5, and F6 (Fig. 6C), indicating that FOXA1 interacts with multiple regions of

**Fig. 4 | BPTF promotes AR binding at promoters, enhancers and super-enhancers of AR target genes. A** Differential analysis of AR ChIP-seq peaks in BPTF-KD Rv1 cells. Peaks enriched in control and BPTF-KD cells are highlighted in red ($p < 0.05$). **B** Heatmap showing decreased AR peak signals in BPTF-KD cells; these peaks are defined as BPTF-dependent AR peaks. **C** Proportion of BPTF-dependent AR peaks localized at active enhancer and promoter regions. **D** Peak profile plots of BPTF-dependent AR peaks, showing AR (top) and BPTF (bottom) signal intensities at active enhancer and promoter regions after BPTF KD in Rv1 cells. **E** Track images showing BPTF and AR peaks at promoter and enhancer regions of representative AR target genes after BPTF KD in Rv1 cells. **F** ChIP-qPCR analysis of AR at the promoter or enhancer regions of representative AR target genes after BPTF KD in Rv1 cells. **G** BETA analysis showing the correlation between BPTF-dependent AR peaks (i.e., AR ChIP-seq peaks reduced upon BPTF KD) and BPTF-activated genes (i.e., transcripts downregulated in BPTF-KD RNA-seq). **H** GO

analysis of direct BPTF-dependent AR target genes identified in (**G**). **I** Identification of super-enhancers (SEs) in Rv1 cells using the rank ordering of super-enhancers (ROSE) algorithm, based on enhancers defined by the H3K27ac/H3K4me1 ChIP-seq peaks. **J** Peak profile plots showing BPTF (left) and AR (right) signal intensities at SE sites after BPTF KD in Rv1 cells. **K** Track images showing BPTF and AR peaks at SEs of representative AR target genes after BPTF KD in Rv1 cells. **L** Dot plot showing the ssGSEA-derived scores for SE-associated genes (n = 80) based on BPTF-KD RNA-seq data. **M** GO analysis of the reduced SE-associated genes after BPTF KD in Rv1 cells. CUT&RUN ChIP-seq data (**A–E, I–K**) were generated from two biological replicates. Data are from three independent biological replicates (**L**) or are representative of three independent biological replicates (**F**). Data are presented as mean ± SD (**F**). Statistical significance was determined using a two-tailed unpaired Student's t-test (**F, L**). Source data are provided as a Source data file.

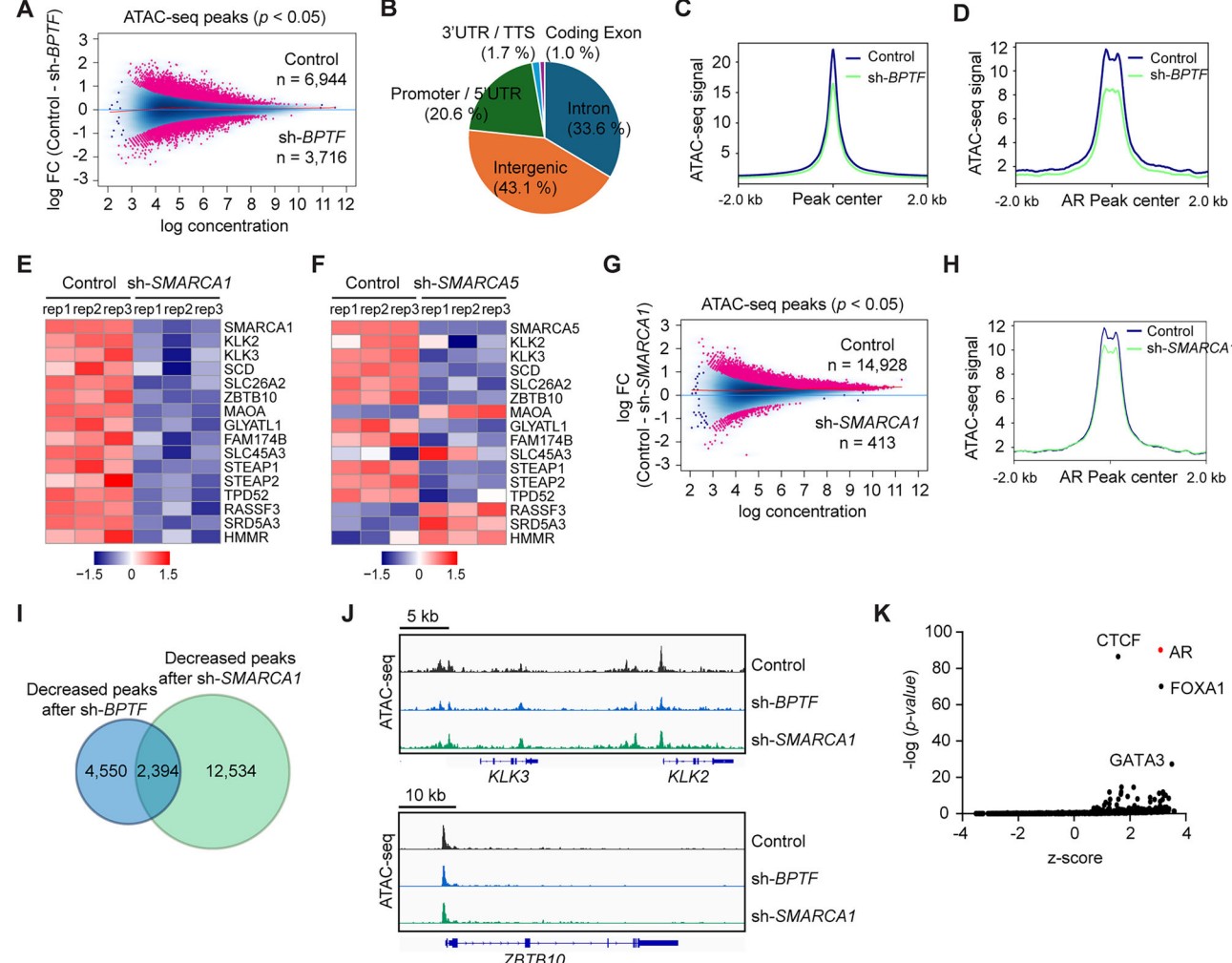

**Fig. 5 | BPTF enhances chromatin accessibility to facilitate AR binding. A** ATAC-seq analysis showing the differential chromatin accessibility between control and BPTF-KD Rv1 cells. **B** Genomic distribution of total ATAC-seq peaks in Rv1 cells. **C** Peak profile plots showing a moderate reduction in global ATAC-seq signal intensity following BPTF KD in Rv1 cells. **D** Peak profile plots showing that BPTF KD reduces ATAC-seq signal intensity at BPTF-dependent AR-binding sites. RNA-seq analysis showing the effect of SMARCA1 (**E**) or SMARCA5 (**F**) knockdown on the expression of a subset of BPTF-dependent AR target genes. **G** ATAC-seq analysis showing the differential chromatin accessibility between control and SMARCA1-KD Rv1 cells. **H** Peak profile plots showing that SMARCA1 KD decreases ATAC-seq signal

intensity at BPTF-dependent AR-binding sites. **I** Overlap of decreased ATAC-seq peaks (2394) between BPTF-KD and SMARCA1-KD Rv1 cells, revealing that 34.5% (2394/6944) of BPTF-dependent peaks are also SMARCA1-dependent. **J** Track images showing reduced ATAC-seq signal at representative AR target genes (*KLK3*, *KLK2*, *ZBTB10*) upon BPTF or SMARCA1 KD. ZBTB10 is an experimentally validated AR target gene[50]. **K** BART analysis of genes associated with 2394 shared ATAC-seq peaks, identifying AR as the top transcriptional regulator of these genes. ATAC-seq data (**A–D, G–K**) were generated from two biological replicates. RNA-seq data were generated from three biological replicates (**E, F**). Source data are provided as a Source data file.

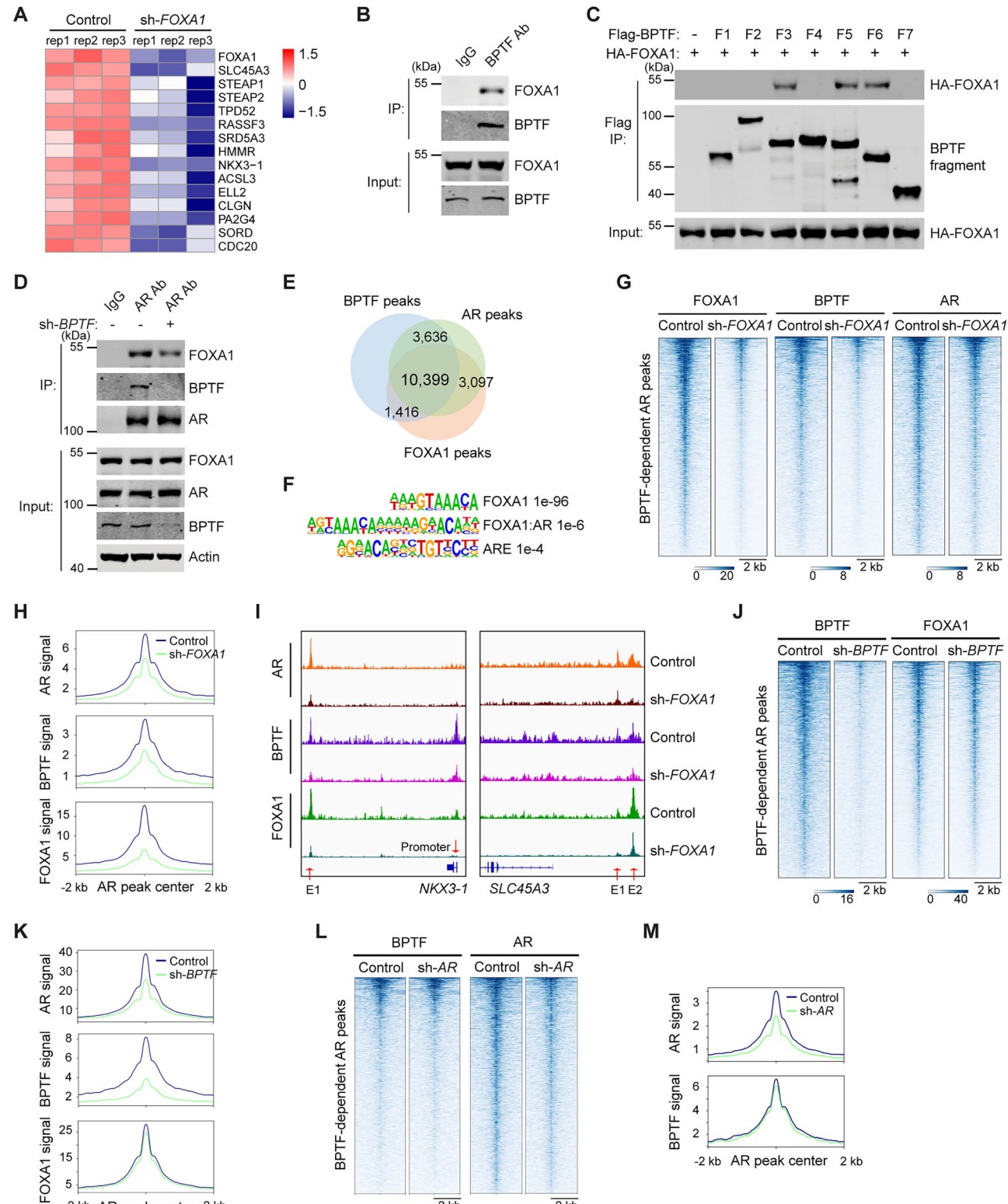

**Fig. 6 | BPTF interacts with FOXA1 and stabilizes AR-FOXA1 interaction on chromatin. A** Heatmap of RNA-seq data showing that FOXA1 KD in Rv1 cells decreases the expression of a subset of BPTF-dependent AR target genes. **B** Co-IP of FOXA1 with BPTF in Rv1 cells. **C** Co-IP of HA-FOXA1 with Flag-tagged BPTF fragments (F3, F5, and F6) in 293 T cells. **D** Reduced co-IP of FOXA1 with AR following BPTF KD in Rv1 cells. **E** Venn diagram showing the shared ChIP-seq peaks of BPTF, AR and FOXA1. **F** HOMER motif analysis showing enrichment of FOXA1 and ARE motifs within the shared ChIP-seq peaks of BPTF, AR, and FOXA1. Heatmap (**G**) and profile plots (**H**) showing that FOXA1 KD in Rv1 cells reduces signal intensities of FOXA1, BPTF, and AR peaks at BPTF-dependent AR-binding sites. **I** Track images showing reduced FOXA1, BPTF, and AR peak signals at the representative AR target genes upon FOXA1 KD in Rv1 cells. Heatmap (**J**) and profile plots (**K**) showing that BPTF KD in Rv1 cells has minimal impact on FOXA1 peak intensity at BPTF-dependent AR-binding sites. Heatmap (**L**) and profile plots (**M**) showing that AR KD has little effect on BPTF peak intensity at BPTF-dependent AR-binding sites. RNA-seq data were generated from three biological replicates (**A**). CUT&RUN ChIP-seq data (**E**–**M**) were generated from two biological replicates. Data are representative of three independent biological replicates (**B**–**D**). Source data are provided as a Source data file.

BPTF. Notably, BPTF KD reduced the co-IP of FOXA1 with AR in Rv1 and C4-2 cells (Fig. 6D, Fig. S6D), suggesting that BPTF enhances the AR-FOXA1 interaction.

To explore the potential FOXA1/BPTF/AR complex on chromatin, we conducted FOXA1 CUT&RUN ChIP-seq for comparison with ChIP-seq of BPTF and AR. Comparative analysis identified 10,399 shared peaks among FOXA1, BPTF, and AR, which accounted for 74.1% of BPTF-AR, 77.1% of FOXA1-AR, and 88.0% of BPTF-FOXA1 co-occupied sites (Fig. 6E). Notably, FOXA1 motifs were most enriched in these shared peaks, whereas AR motifs were weakly enriched (Fig. 6F, Fig. S6E). To evaluate the effect of FOXA1 on chromatin binding of AR and BPTF, we performed CUT&RUN ChIP-seq of FOXA1, BPTF, and AR in FOXA1-KD Rv1 cells. FOXA1 KD reduced the occupancy of FOXA1, BPTF, and AR at BPTF-dependent AR-binding sites, as shown by peak signal reductions (Fig. 6G, H), representative track images (Fig. 6I), and ChIP-qPCR analyses at the *NKX3-1* and *SLC45A3* loci in Rv1 (Fig. S6F–H) and VCaP cells (Fig. S6I–K). These results indicate that FOXA1 promotes the chromatin binding of both BPTF and AR. To investigate the effect of BPTF on FOXA1 chromatin binding, we performed FOXA1 CUT&RUN ChIP-seq on BPTF-KD Rv1 cells. BPTF KD only slightly reduced the signal of FOXA1 peaks at BPTF-dependent AR-binding sites (Fig. 6J, K), suggesting that BPTF may not significantly regulate FOXA1 chromatin binding. In contrast, BPTF KD significantly reduced the signal of AR peaks at these sites (Figs. 4B, 6K). This indicates that BPTF enhances AR binding to FOXA1 on chromatin, consistent with the observation that BPTF KD reduced the co-IP between AR and FOXA1 (Fig. 6D, Fig. S6D). Finally, to evaluate the effect of AR on BPTF chromatin binding, we performed BPTF CUT&RUN ChIP-seq on AR-KD Rv1 cells. AR KD had little effect on the signal of BPTF peaks at BPTF-dependent AR-binding sites (Fig. 6L, M). Together, these results suggest that FOXA1 recruits AR-BPTF, while BPTF stabilizes the AR-FOXA1 interaction at BPTF-dependent AR-binding sites.

### BPTF is clinically relevant to PCa progression

To assess the clinical relevance of BPTF, we analyzed the association between *BPTF* expression or a BPTF-dependent gene signature and PCa patient outcomes. Although high *BPTF* expression showed a trend toward reduced overall survival in a GEO dataset, the difference did not reach statistical significance, likely due to limited sample size ($p = 0.115$; Fig. 7A). In contrast, elevated expression of the BPTF gene signature, defined as the top 10 genes most significantly decreased in BPTF-KD RNA-seq, was associated with shorter disease-free survival in the TCGA PCa dataset (Fig. 7B). These findings suggest that BPTF and its downstream gene signature may serve as potential prognostic indicators in PCa.

To investigate the clinical relevance of BPTF-dependent AR chromatin binding, we analyzed publicly available AR ChIP-seq data from 18 primary PCa samples and 15 CRPC patient-derived xenograft (CRPC-PDX) samples, focusing on the top 2000 BPTF-dependent AR peaks. Principal component analysis (PCA) revealed that these peaks clearly separate primary PCa and CRPC-PDX samples into distinct clusters (Fig. 7C), reflecting differences in AR peak signal intensity (Fig. 7D, E). Unsupervised hierarchical clustering further stratified the CRPC-PDX samples into two subgroups (CRPC-1 and CRPC-2) based on AR peak signal intensity, which was high in CRPC-2 and moderate in CRPC-1 (Fig. 7F–H). Using RNA-seq data from the same CRPC-PDX samples, we found that AR activity, measured by AR scores derived from a 20-gene AR signature, was significantly higher in CRPC-2 compared to CRPC-1 (Fig. 7K). Notably, *BPTF* expression, but not *AR* expression, was elevated in CRPC-2 relative to CRPC-1 (Fig. 7I, J), suggesting that high *BPTF* expression may contribute to the enhanced AR chromatin binding and transcriptional activity in CRPC-2. These findings indicate that BPTF-dependent AR peaks may help classify PCa subtypes with distinct AR activity and further support a clinically relevant role for BPTF in PCa progression.

### BPTF inhibitor can inhibit AR activity and PCa cell growth

The BPTF-AR interaction is mediated by two domains: BPTF-BRD and AR-LBD (Fig. 3B–D). To predict the 3D interaction model of the two domains, we used Alphfold 2. The results suggest that the BPTF-BRD pocket region interacts with the co-factor binding groove of AR-LBD, which consists of Helix 4, 5, and 12 (Fig. 8A, B, Fig. S7A). AU1 is a selective BPTF inhibitor that targets the BRD pocket[39]. Notably, AU1 has been reported to inhibit targets beyond BPTF, and therefore off-target effects cannot be entirely excluded[40]. Molecular docking simulations of AU1 showed that it localizes to the interface between the BPTF-BRD and AR-LBD (Fig. 8C), suggesting that AU1 could disrupt the BPTF-AR interaction. AU1 treatment for 24 h had no effect on the protein levels of BPTF, AR, and FOXA1 in PCa cells (Fig. 8D, Fig. S7B), but it reduced the co-IP of AR with BPTF in Rv1 and C4-2 cells (Fig. 8E). Treatment with increasing concentrations of AU1 (0, 5, and 10 μM) for 24 h gradually decreased the mRNA levels of several AR target genes in PCa cells (Fig. 8F, Fig. S7C). To assess the effect of AU1 on PCa cell growth, we performed a colony formation assay using escalating doses of AU1, administered every 3 days, to determine the $IC_{50}$. The $IC_{50}$ values of AU1 for various PCa cell lines ranged from 2.3 to 3.6 μM (Fig. 8G, H, Fig. S7D). Enzalutamide (ENZ, 5 μM) inhibited colony formation of androgen-sensitive LNCaP and VCaP cells by ~43% and ~20%, respectively, but had less effect on androgen-insensitive C4-2 cells (~16% inhibition) and no effect on enzalutamide-resistant Rv1 cells (Fig. 8I, J). In contrast, AU1 ($IC_{50}$ concentration) inhibited colony formation of all PCa cell lines by ~50% (Fig. 8I, J). Notably, the combination of enzalutamide (5 μM) and AU1 ($IC_{50}$ concentration) synergistically inhibited Rv1, C4-2, and VCaP cells, reducing colony formation by ~90% (Fig. 8I, J). These results indicate that AU1 can sensitize these PCa cells to enzalutamide, even at a sub-$IC_{50}$ concentration that is ineffective on its own. Together, these results indicate that the BPTF inhibitor AU1 can effectively inhibit PCa cell growth either alone or in combination with enzalutamide.

## Discussion

BPTF acts as an oncogenic protein in various cancer types, but its role in PCa remains unclear. We found that BPTF is upregulated in CRPC compared to primary PCa. BPTF-dependent gene signature in PCa is associated with poorer patient outcomes. Functionally, BPTF inhibition suppresses PCa cell proliferation, while BPTF overexpression promotes androgen-independent growth. These findings demonstrate that BPTF is a key driver of PCa progression. Although BPTF has previously been reported to promote the progression of multiple cancer types by regulating c-Myc activity, our study shows that BPTF primarily upregulates the AR transcriptional program in PCa. BPTF promotes AR binding to enhancers, super-enhancers, and promoters of AR target genes, resulting in increased AR target gene expression. *BPTF* mRNA levels positively correlate with an AR target gene signature in the TCGA PCa dataset. Moreover, BPTF-dependent AR ChIP-seq peaks are positively associated with both *BPTF* expression and AR activity in a cohort of CRPC-PDX samples. Together, these findings support a model in which BPTF promotes CRPC progression by enhancing AR activity. We identified two mechanisms by which BPTF facilitates AR function: (1) increasing chromatin accessibility to promote AR binding, and (2) stabilizing the interaction between AR and FOXA1 on chromatin (Fig. S8A).

Our findings that BPTF enhances chromatin accessibility, in part through SMARCA1, to facilitate AR binding are consistent with several published studies on BAP18, a BPTF-associated protein. BAP18 has been shown to recruit the MLL1 methyltransferase complex to AR, thereby enhancing AR activity in PCa cells[41]. In breast cancer cells, BAP18 interacts with the BPTF-SMARCA1 complex to facilitate CTCF recruitment to ERα, thereby promoting promoter-enhancer looping and ERα target gene expression[42]. In leukemia, BAP18 interacts with the BPTF-SMARCA5 complex to regulate the accessibility of insulator

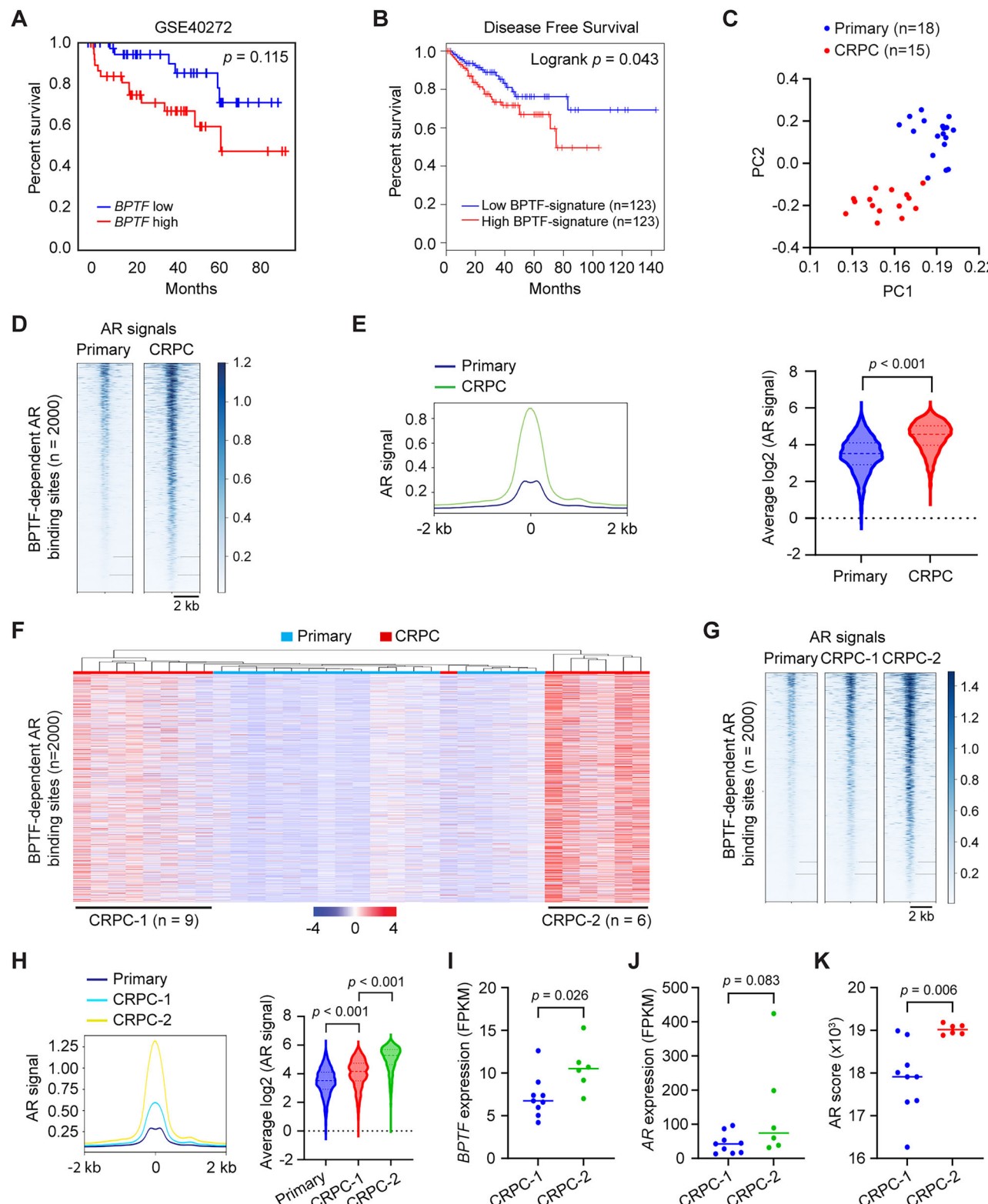

regions, thereby enhancing CTCF binding and boundary formation that are critical for maintaining leukemic transcriptional programs[21]. Notably, CTCF is the top enriched transcription factor motif in our BPTF ChIP-seq peaks (Fig. S3B). These findings raise the intriguing possibility that BPTF and BAP18 may cooperate to recruit CTCF to AR-binding sites and regulate promoter-enhancer looping or topologically associated domains (TADs), providing additional mechanisms for modulating AR activity.

We find that BPTF-BRD interacts with AR-LBD, while multiple BPTF fragments interact with FOXA1. Previous studies have shown that the DBD/hinge region of AR directly interacts with the forkhead domain of FOXA1[43,44]. Thus, multiple interactions among BPTF, AR, and FOXA1 may lead to the formation of a stable complex, with FOXA1 anchoring AR-BPTF to chromatin and BPTF stabilizing the AR-FOXA1 interaction. This model is supported by our observations at BPTF-dependent AR-binding sites: FOXA1 knockdown reduces BPTF and AR peaks, whereas

**Fig. 7 | Clinical relevance of BPTF in prostate cancer. A** Kaplan–Meier plot showing the association between *BPTF* expression and overall survival in PCa patients. The high *BPTF* expression tends to correlate with shorter survival ($p = 0.115$). **B** Kaplan–Meier plot showing the association between BPFF-dependent gene signature and disease-free survival in the TCGA PCa dataset. A high BPTF signature is associated with shorter survival. **C** Principal component analysis (PCA) of AR peak intensities at the top 2000 BPTF-dependent AR-binding sites in primary PCa (n = 18) and CRPC-PDX (n = 15) samples. Each dot represents the averaged AR peak signal for an individual sample. Heatmap (**D**) and profile plot (**E**) showing the averaged AR signal intensity at the top 2000 BPTF-dependent AR-binding sites in primary PCa and CRPC-PDX samples, with corresponding quantification shown on the right. **F** Unsupervised hierarchical clustering of primary PCa and CRPC-PDX samples based on the top 2000 BPTF-dependent AR peaks identifies three distinct groups: primary PCa (low AR signal), CRPC−1 (moderate AR signal), and CRPC-2 (high AR signal). Heatmap (**G**) and profile plot (**H**) showing the averaged AR signal intensity at the top 2000 BPTF-dependent AR-binding sites across primary PCa, CRPC−1, and CRPC-2 samples, with corresponding quantification shown on the right. RNA-seq data from CRPC-PDX samples (GSE130408) showing that CRPC-2 samples exhibit increased *BPTF* mRNA expression (**I**) and AR activity score (**K**), but not *AR* mRNA expression (**J**). Statistical significance was determined using a two-tailed unpaired Student's t-test (**E, I–K**); One-way ANOVA (two-sided) with Tukey's multiple comparison test (**H**). Source data are provided as a Source data file.

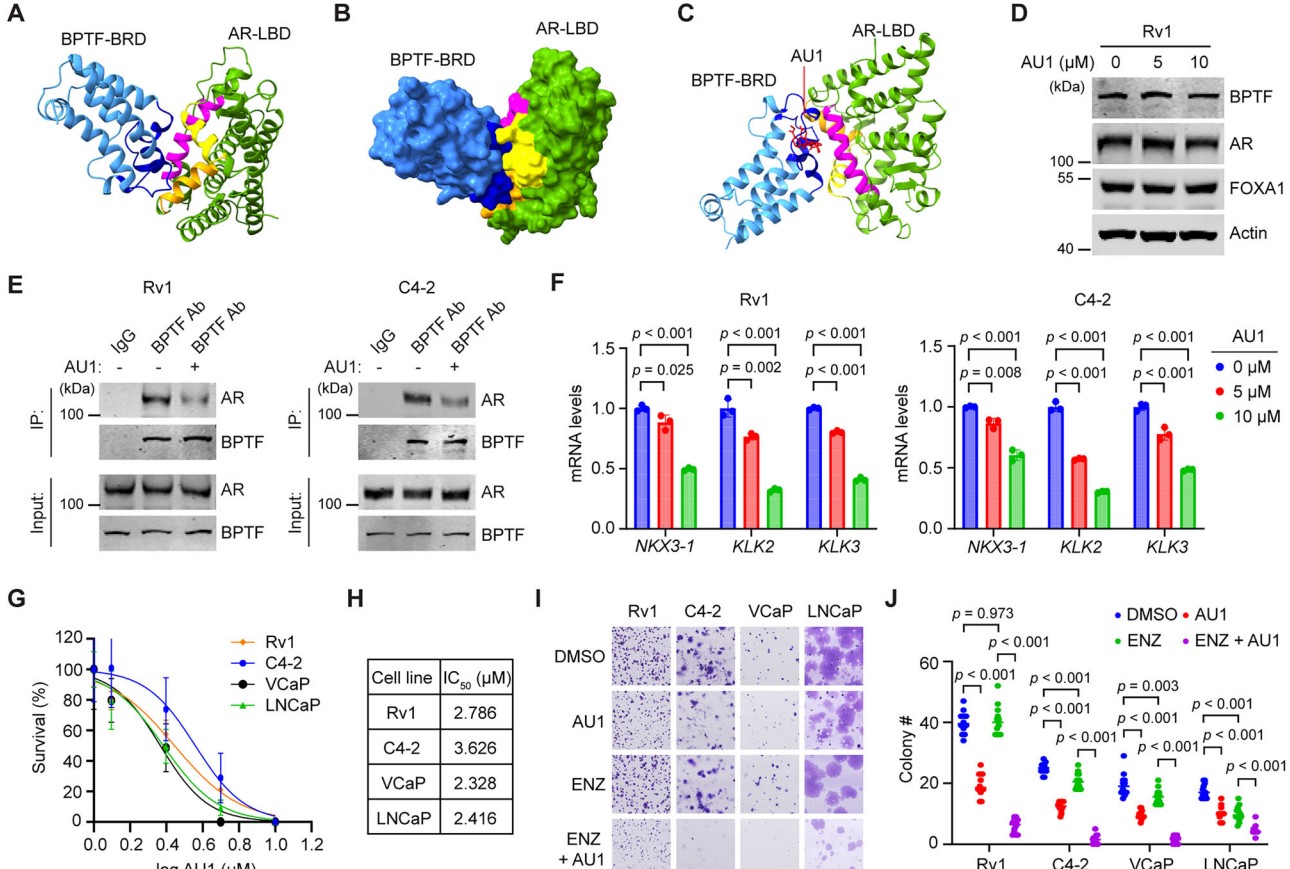

**Fig. 8 | The BPTF inhibitor AU1 disrupts the AR-BPTF interaction, suppresses AR target gene expression, and inhibits PCa cell growth. A, B** Predicted 3D interaction model of BPTF-BRD (cyan) and AR-LBD (green) generated by Alpha-Fold2. The BPTF-BRD pocket region is highlighted in dark blue, while H3, H4, and H12 of the AR co-factor binding groove are highlighted in magenta, orange, and yellow, respectively. The model is visualized as a ribbon diagram (**A**) or a solid surface (**B**). **C** Superimposed structure of AU1-docked BPTF-BRD with the predicted BPTF-BRD and AR-LBD interaction model. AU1 is highlighted in red, the BPTF-BRD pocket in dark blue, and H3, H4, and H12 of the AR co-factor binding groove in magenta, orange, and yellow, respectively. The superimposed model is visualized as a ribbon diagram. **D** Western blot showing BPTF, AR, and FOXA1 protein levels after AU1 treatment in Rv1 cells. **E** Decreased co-IP of AR with BPTF following AU1 treatment in Rv1 and C4-2 cells. **F** qRT-PCR analysis showing that AU1 treatment decreases mRNA levels of representative AR target genes in Rv1 and C4-2 cells. **G, H** IC$_{50}$ determination of AU1 in PCa cells using colony formation assays. Colony formation assays evaluating the effects of DMSO, AU1 (IC$_{50}$ concentration), enzalutamide (ENZ, 5 μM), and combination treatment (ENZ + AU1) on PCa cells. Example images (**I**) and quantification (**J**) of colony formation are shown. Data are representative of three independent biological replicates (**D–J**). Data are presented as mean ± SD (**F, G**). Statistical significance was determined using one-way ANOVA (two-sided) with Tukey's multiple comparison test (**F, J**). Source data are provided as a Source data file.

BPTF knockdown has a minimal effect on FOXA1 peaks but reduces AR peaks. Notably, the regions co-occupied by BPTF, AR, and FOXA1 exhibit strong enrichment of FOXA1 motifs but weak enrichment of AR motifs (Fig. 6F, Fig. S6E). FOXA1 has been reported to recruit AR to low-affinity half-AREs[37]. Thus, BPTF may play a key role in the binding of AR to chromatin sites with strong FOXA1 motifs but weak AR motifs, acting as a bridging and stabilizing factor for FOXA1-AR interactions.

BPTF's interactions with and regulation of AR and ERα support the idea that BPTF promotes lineage-specific transcriptional programs through distinct mechanisms, rather than simply acting as a general amplifier of existing lineage identity programs. The observations that BPTF significantly enhances AR chromatin binding while having minimal effect on FOXA1 binding at the shared genomic loci further suggest that BPTF does not uniformly enhance the binding of

lineage-specific transcription factors but instead employs distinct mechanisms to regulate their activity.

SMARCA1 and SMARCA5 are alternative catalytic subunits of the NURF complex[8]. Our findings indicate that SMARCA1 and SMARCA5 exert both overlapping and distinct effects on the expression of BPTF-dependent AR target genes, and they do not appear to act redundantly in regulating their shared AR targets. These results suggest that SMARCA1 and SMARCA5 may define distinct NURF complexes, each capable of co-regulating a subset of AR target genes while differentially modulating others. This intriguing possibility warrants further investigation.

Structural modeling suggests that the bromodomain (BRD) pocket of BPTF interacts with the co-factor binding groove of the AR ligand-binding domain (LBD), an interaction that can be targeted by the BPTF inhibitor AU1. Consistently, AU1 treatment suppresses AR activity and PCa cell growth, both as a monotherapy and in combination with AR pathway inhibitors. Furthermore, AR antagonists may inhibit the AR-BPTF interaction by altering the conformation of the AR-LBD co-factor binding groove. These findings provide proof-of-concept that targeting the BPTF-AR interaction is a viable strategy to repress AR activity and inhibit PCa cell growth.

We demonstrate that BPTF upregulates the expression of a subset of AR hallmark genes through SMARCA1 and/or FOXA1 (Fig. S8B). However, some BPTF-dependent AR target genes are independent of either SMARCA1 or FOXA1 (Fig. S8B), suggesting that BPTF may employ additional mechanisms to regulate AR activity. Our RNA-seq, ChIP-seq, and ATAC-seq analyses suggest that BPTF may regulate additional transcription factors and signaling pathways beyond AR signaling. Future works will be necessary to determine whether BPTF regulates additional transcription factors, contributing to both AR-dependent and AR-independent transcriptional programs that drive PCa progression.

In summary, we have identified a crucial role for BPTF in modulating AR activity and PCa cell growth. These findings provide new potential targets for prostate cancer therapy.

# Methods

## Antibodies and reagents
Antibodies were obtained from the following sources: BPTF (ABE24, ABE1966, MABE443) and AR (06-680) from EMD Millipore (Burlington, MA); c-Myc (#9402) and HA-tag (#3724) from Cell Signaling Technology (Danvers, MA); SMARCA1 (13-2005), FOXA1 (13-2001), H3K27ac (13-0059), H3K4me1 (13-0040), and Rabbit IgG control (13-0042) from EpiCypher (Durham, NC); Myc (sc-40) from Santa Cruz Biotechnology (Dallas, TX); Flag (F7425, F3165) and actin (A5441) from Sigma-Aldrich (St. Louis, MO); Cas9 (61957) and RNA Pol II (39497) from Active Motif (Carlsbad, CA); Mouse IgG control (02-6300) and fluorescent secondary antibodies (A-21109, A-21058, A-11375) from Invitrogen (Waltham, MA); and fluorescent TrueBlot secondary antibodies (18-4416-32, 18-4417-32) from Rockland Immunochemicals (Pottstown, PA). The amounts or dilutions of antibodies used in the experiments are listed in Supplementary Data 3. Enzalutamide was purchased from Abmole Bioscience (Kowloon, Hong Kong), R1881 from Sigma-Aldrich (St. Louis, MO), and GSK1379725A (AU1) from AOBIOUS (Gloucester, MA). All reagents were used according to the manufacturers' instructions.

## Cell lines
Rv1 cells (also called CWR22Rv1 cells) were provided by Dr. James Jacobberger (Case Western Reserve University, Cleveland, Ohio). C4-2 cells were provided by Dr. Leland Chung (Cedars-Sinai Medical Center, Los Angeles, CA). LNCaP, VCaP, and SW620 cells were purchased from the American Type Culture Collection (ATCC). Rv1, C4-2, and LNCaP cells were maintained in RPMI 1640 media supplemented with 10% FBS and antibiotics. VCaP cells were maintained in DMEM media

supplemented with 10% FBS and antibiotics. SW620 cells were maintained in RPMI 1640 media supplemented with 10% FBS, 1% L-glutamine, and antibiotics. Cells were periodically checked for Mycoplasma by PCR analysis, and cells of <20 passages were used for experiments.

## Plasmids
HA-AR (WT: 1-919) and Flag-AR (WT: 1-919; N-TAD: 1-537; DBD: 537-669; LBD: 669-919) were cloned into pcDNA3.1 vector[25,45]. pFastBac1_FLAG-BPTF plasmid, encoding human BPTF (amino acids 140−2903), was a gift from Joe Landry (Addgene plasmid # 102242). The BPTF plasmid was used as template in PCR to generate BPTF truncation mutants and cloned into Flag-pcDNA3.1 vector including F1 (140-529), F2 (521-1010), F3 (1002-1389), F4 (1390-1794), F5 (1786-2254), F6 (2246-2626), F7 (2618-2903), F6 + 7 (2246-2903) and F6 + 7-BRD (2246-2781). BPTF fragment F7 (2618-2903) was cloned into myc-pcDNA3.1 vector. Flag-FOXA1 was a gift from Stefan Koch (Addgene plasmid # 153109), and HA-FOXA1 was cloned into pcDNA3.1 vector. LentiMPHv2 (Addgene plasmid # 89308) and lentiSAMv2 (Addgene plasmid # 75112) were gift from Feng Zhang. shRNAs in pLKO.1 vector targeting *BPTF* (TRCN0000016819, TRCN0000319225), *FOXA1* (TRCN0000014879), *SMARCA1* (TRCN0000303644), *SMARCA5* (TRCN0000013217) and *AR* (TRCN0000003717) were from Sigma-Aldrich (St. Louis, MO). The control pLKO.1 vector harbors a non-targeting shRNA (TCTCGCTTGGGCGAGAGTAAG). Plasmids were confirmed by Sanger sequencing at Genewiz (South Plainfield, NJ).

## BPTF staining in PCa tissue microarrays (TMAs)
Commercial PCa TMAs were purchased from the Molecular Pathology Core of Vancouver Prostate Centre at the University of British Columbia. 4-μm-thick sections of PCa TMAs were used for immuno-histochemistry staining of BPTF, and staining procedures were detailed previously[46]. Briefly, antigen retrieval was performed using Dako target retrieval solution, followed by a 3% hydrogen peroxidase block for 30 min. Specimens were incubated with BPTF antibody (MABE443) diluted in Dako antibody diluent (1:300) overnight at 4 °C. Slides were then washed three times with PBS/Tween-20 and incubated with Dako-labeled polymer-HRP (anti-mouse) for 1 h at room temperature. Slides were then washed four times with PBS/Tween-20, developed with DAB, and counterstained with hematoxylin. Stained TMA slides were digitized with Leica scanner (Aperio AT2, Leica Microsystems; Concord, Ontario, Canada) at magnification equivalent to 40X. The images were subsequently stored in the Aperio eSlide Manager (Leica Microsystems) at the Vancouver Prostate Centre. The staining intensity was scored as 0 (none), 1 (weak), 2 (moderate) or 3 (strong). The percentage of cancer cells with positive staining was also scored. The BPTF H-score was calculated using the formula: staining intensity x percentage of positive cancer cells. The H-score of different PCa groups was compared using the Kruskal−Wallis test or the Mann−Whitney test for statistical analysis.

## Lentiviral vector packaging and transduction
Lentiviral vectors were packaged in 293 T cells using calcium phosphate transfection. The supernatant containing lentiviral particles were collected 48 h after transfection. PCa cells were then transduced with supernatants in the presence of polybrene (8 μg/ml) for 24 h before replacement with fresh growth media. Cells were analyzed 48 h post-transduction.

## Immunoprecipitation (IP) and western blotting
For IP, cells were lysed in buffer containing 50 mM Tris-HCl, pH 7.5, 150 mM NaCl, 0.5% NP-40, 1 mM EDTA, 1 mM sodium orthovanadate, and 1× protease inhibitor cocktail. For IP of Flag-tagged proteins, lysates were incubated overnight with M2 beads (Sigma-Aldrich, St. Louis, MO), followed by three washes and elution with SDS loading buffer. For endogenous protein IP, lysates were incubated overnight

with 3 μg of rabbit primary antibody, then for 4 h with TrueBlot anti-rabbit IgG beads. After three washes, proteins were eluted with SDS loading buffer. Whole cell lysates were prepared using RIPA buffer (50 mM Tris-HCl, pH 7.5, 150 mM NaCl, 1% Triton X-100, 0.1% SDS, 0.1% sodium deoxycholate, 1 mM EDTA, 1 mM sodium orthovanadate, and 1× protease inhibitor cocktail). Protein samples were resolved by SDS-PAGE and transferred to nitrocellulose membranes. For IP samples, membranes were probed with primary antibodies followed by fluorescent TrueBlot secondary antibodies. For whole cell lysates, blots were incubated with primary antibodies followed by fluorescent secondary antibodies. All blots were imaged using the Odyssey imaging system (LI-COR Biotechnology, Lincoln, NE).

### Rapid immunoprecipitation mass spectrometry of endogenous proteins (RIME)

RIME was performed as previously described with minor modifications[47]. Briefly, $6 \times 10^7$ Rv1 cells were crosslinked with 1% formaldehyde for 10 min at room temperature and quenched with 100 mM glycine. Nuclei were isolated using lysis buffer (50 mM HEPES-KOH, pH 7.5, 140 mM NaCl, 1 mM EDTA, 10% glycerol, 0.5% NP-0.5, 0.25% Triton X-100, and protease inhibitor cocktail). The nuclear pellet was resuspended in shearing buffer (10 mM Tris-HCl, pH 7.6, 1 mM EDTA, 0.5 mM EGTA, 0.1% SDS), and chromatin was sheared using a Covaris M220 Focused-ultrasonicator for 4 min. Sheared chromatin was diluted 10-fold in dilution buffer (20 mM Tris-HCl, pH 8.0, 150 mM NaCl, 2 mM EDTA, 1% Triton X-100, 0.01% SDS) and incubated overnight at 4 °C with 10 μg of AR antibody or control IgG pre-bound to 100 μL of Protein A/G beads. Beads were washed 10 times with RIPA buffer and processed for on-bead trypsin digestion and LC-MS analysis (Proteomics Core, Sanford Burnham Prebys, La Jolla).

### qRT-PCR analysis

Total RNA from cells was prepared using a total RNA miniprep kit (Sigma-Aldrich, St. Louis, MO). cDNA was synthesized using random hexamers. SYBR Green qPCR analysis was performed using an Mx3005P QPCR system (Agilent Technologies). Primers for peptidyl-prolyl isomerase A (PPIA) served as an internal control. Primers for qPCR analysis of human gene transcripts were: *PPIA*: 5′-GACCCAACA-CAAATGGTTC-3′, 5′-AGTCAGCAATGGTGATCTTC-3′; *BPTF*: 5′-CTTCA GGAGCCATAGTACCTACA-3′, 5′-CAAGGGGCGGGATGTCTTTTT-3′; *AR*: 5′-CCATCTTGTCGTCTTCGGAAATGTTATGAAGC-3′, 5′-AGCTTCTGGG TTGTCTCCTCAGTGG-3′; *FOXA1*: 5′-GCAATACTCGCCTTACGGCT-3′, 5′-TACACACCTTGGTAGTACGCC-3′; *KLK2*: 5′-GGTCGGCACAACCTGTTT GA-3′, 5′-GCCCAGGACCTTCACAACAT-3′; *KLK3*: 5′-ACCAGAGGAGTT CTTGACCCCAAA-3′, 5′-CCCCAGAATCACCCGAGCAG-3′; *NKX3-1*: 5′-ACTTGGGGTCTTATCTGTTGGA-3′, 5′-CTCGATCACCTGAGTGTGGG-3′; *SLC45A3*: 5′-GACACTATGATGAAGGCGTTCG-3′, 5′-GAGAAGGTGAA CCCGGTGAG-3′; *TMPRSS2*: 5′-CCTCTAACTGGTGTGATGGCGT-3′, 5′-TGCCAGGACTTCCTCTGAGATG-3′; *c-Myc*: 5′-GGCTCCTGGCAAAAGG TCA-3′, 5′-AGTTGTGCTGATGTGTGGAGA-3′; *SMARCA1*: 5′-GCTGGAGA CTACCGCCATAG-3′, 5′- CAACCAATTCAGTCCTCGAATCT-3′; *SMARCA5*: 5′-TGCAAACTGACCGGGCAAATA-3′, 5′-TCGCCAACGGA TAGTAAGTTCT-3′; *STEAP2*: 5′-GGTCACTGTAGGTGTGATTGG-3′, 5′-ACCACATGATAGCCGCATCTAA-3′; *CCNB1*: 5′-AATAAGGCGAAGATC AACATGGC-3′, 5′-TTTGTTACCAATGTCCCCAAGAG-3′; *CCNE2*: 5′-TCAA GACGAAGTAGCCGTTTAC-3′, 5′-TGACATCCTGGGTAGTTTTCCTC-3′; *CDC20*: 5′-GACCACTCCTAGCAAACCTGG-3′, 5′-GGGCGTCTGGCTGTT TTCA-3′; *MCM3*: 5′-GGCCTCCATTGATGCTACCTA-3′, 5′-ACTTTGG-GACGAACTAGAGAACA-3′; *MKI67*: 5′-ACGCCTGGTTACTATCAAAAGG-3′, 5′-CAGACCCATTTACTTGTGTTGGA-3′.

### ChIP-qPCR

$5 \times 10^6$ cells were crosslinked 10 min in 1% formaldehyde at room temperature and then quenched with 125 mM glycine. Cell nuclei were extracted with lysis buffer (10 mM Tris-HCl, pH 7.5, 10 mM NaCl, 0.2%

NP-40, protease inhibitor cocktail). Nuclear pellets were resuspended in 0.6 ml MNase digestion buffer (20 mM Tris-HCl, pH 7.5, 15 mM NaCl, 60 mM KCl, 1 mM CaCl2, and a protease inhibitor cocktail) followed by MNase (20,000 U) digestion at 37 °C for 20 min. The reaction was stopped by adding 2X Stop/ChIP buffer (100 mM Tris-HCl, pH8.1, 20 mM EDTA, 200 mM NaCl, 2% Triton X-100, 0.2% sodium deoxycholate). Chromatin samples were incubated with 3 μg primary antibody or control IgG overnight at 4 °C. Protein A/G beads were added and incubated at 4 °C for 3 h. After four washes, bound chromatin was eluted from beads for reverse crosslinking and digestion with RNase A and proteinase K. DNA was purified with spin columns and subjected to qPCR analysis. ChIP-qPCR was performed in biological triplicates, and independent experiments were repeated at least three times. Data were presented as the percentage of input. PCR primers for ChIP-qPCR were:

*KLK3* enhancer: 5′-TGGGACAACTTGCAAACCTG-3′, 5′-CCA-GAGTAGGTCTGTTTTCAATCCA-3′; *KLK2* promoter: 5′-GGGAATGCCT CCAGACTGAT-3′, 5′-CTTGCCCTGTTGGCACCTA-3′; *NKX3-1* promoter: 5′-CCTCTGGCTCTGGCTCTG-3′, 5′-GTCCTTCCTCATCCAGGACA-3′; *NKX3-1* enhancer1: 5′-GGGCTCACAGTGCTTTAGGA-3′, 5′-GAATCATGC TGACCCTGTTCT-3′; *SLC45A3* enhancer1: 5′-CCCACCGTCATTCTTGC TTT-3′, 5′-TGCTTGTCTATCCAGTGCCA-3′; *SLC45A3* enhancer2: 5′-ACT CTGACTCTGCTAAGCCC-3′, 5′-TCCCTTGGTGCAGATGACTT-3′; *BPTF* promoter: 5′-ACTCGACGCTCCCGCGCTC-3′, 5′-GCCGCCATCTTGTTT CTTC-3′.

### CRISPRa synergistic activation mediator (SAM)

The following oligos were synthesized and cloned into the lentiSAMv2 plasmid: sgRNA-1 targeting *BPTF* promoter (5′-CACCGGCGCGG-GAGCGTCGAGTCGG-3′, 5′-AAACCCGACTCGACGCTCCCGCGCC-3′), sgRNA-2 targeting BPTF promoter (5′-CACCGGAGCCTGGGACG-GAGCGAAG-3′, 5′-AAACCTTCGCTCCGTCCCAGGCTCC-3′), and non-targeting control sgRNA (5′-CACCGGTATTACTGATATTGGTGGG-3′, 5′-AAACCCCACCAATATCAGTAATACC-3′). Rv1 cells were transduced with lentivirus carrying LentiMPHv2 constructs. After 48 h, cells were treated with Hygromycin B (1 mg/ml) and cultured for 7 days to remove non-transduced cells and establish stable cells. The stable cells were transduced with lentivirus carrying LentiSAMv2 constructs (Control sgRNA, BPTF sgRNA-1, or BPTF sgRNA-2). After 48 h, cells were treated with Blasticidin (50 μg/ml) and cultured for 7 days to remove non-transduced cells.

### Colony formation assay

PCa cells were seeded at low density into 6-well plates in triplicate. Cells were treated with AU1 and/or enzalutamide 24 h after seeding, and fresh inhibitors were replenished every 3 days. After 2 weeks, cells were fixed in 3.5% paraformaldehyde and stained with 0.2% crystal violet. Cell colony images were taken with a scanner. The number of colonies (>100 μm in diameter) was determined in 12 higher-power fields.

### RNA-sequencing (RNA-seq)

Rv1 cells were transduced with sh-Control, sh-*BPTF*, sh-*SMARCA1*, sh-*SMARCA5*, or sh-*FOXA1* lentivirus for 48 h. Total RNAs were extracted using a total RNA miniprep kit (Sigma-Aldrich, St. Louis, MO). Three biological replicates of RNA samples were used for library preparation and sequencing (PE150, 20 million reads per library) by Novogene Corporation (Sacramento, CA). Sequenced reads with low quality or adapter contamination were removed. Clear reads were mapped to the *Homo sapiens* reference genome (hg19) using Hisat2 v2.2.1 software. To count the reads numbers mapped to each gene, featureCounts v2.0.6 was used. The fragments per kilobase of transcript per million mapped reads (FPKM) method was used to estimate expression levels. Differential expression analysis was performed using the DESeq2 R package (1.42.0).

## CUT&RUN ChIP-sequencing (ChIP-seq)

Rv1 cells were transduced with sh-Control, sh-BPTF, sh-FOXA1, or sh-AR lentivirus for 48 h. Two biological replicates ($5 \times 10^5$ live cells per replicate) were used for ChIP-seq preparation, following the CUT&RUN kit protocol (EpiCypher). CUT&RUN procedures were detailed previously[48]. Briefly, cell nuclei were extracted and incubated with ConA Beads to absorb nuclei onto beads. Antibody buffer and 0.5 μg of antibody were added to the samples and incubated overnight at 4 °C. After washing beads, 2.5 μl of pAG-NMase was added and incubated for 10 min at room temperature. After washing, 1 μl of chromatin digest additive was added and incubated for 2 h at 4 °C. Stop buffer and 0.5 ng of Spike-in DNA were then added and incubated for 10 min at 37 °C. DNA was purified using the DNA purification kit (EpiCypher). 5 ng of purified DNA was used for library preparation with the Illumina library prep kit (NEB). The library was purified with 1.0× AMPure beads (Beckman) and library fragment size was evaluated using an Agilent Bioanalyzer. Libraries were sequenced (PE150, 20 million reads per library) at Novogene Corporation (Sacramento, CA). The quality of ChIP-seq reads was checked using FastQC software, and Trim Galore was used to remove adapter sequences. After trimming, reads with a score >28 were aligned to the *Homo sapiens* hg19 reference genome and the *E. coli* K12 MG1655 reference genome using Bowtie2. Sequencing data was normalized using a normalization factor calculated from the sequencing depth of *E. coli* Spike-in DNA reads. SAM files were converted to BAM files, and PCR duplicates were removed. Spike-in normalized bigwig and bedgraph files were generated by deepTools. Bedgraph files were used for peak calling with MACS2 bdgpeakcall.

## Assay for transposase accessible chromatin sequencing (ATAC-Seq)

Rv1 cells were transduced with sh-Control, sh-BPTF, or sh-SMARCA1 lentivirus for 48 h. Two biological replicates ($1 \times 10^5$ live cells per replicate) were used for sample preparation, following the ATAC-seq kit manual (Active Motif)[49]. In brief, cell nuclei were extracted using ATAC lysis buffer and incubated with 50 μl of Tagmentation Master Mix at 37 °C for 30 min in a thermomixer. The transposed DNA was purified with spin column and subjected to library preparation using Nextera Index Kit. The library was purified with double-sided SPRI beads, and library fragment size was evaluated using an Agilent Bioanalyzer. Libraries were sequenced (PE150, 100 million reads per library) at Novogene Corporation (Sacramento, CA). Quality of ATAC-seq reads was checked using FastQC software, and Trim Galore was used to remove adapter sequences. After trimming, reads were aligned to the *Homo sapiens* hg19 reference genome using Bowtie2−very-sensitive option. SAM files were converted to BAM files, and mito-chondrial reads and PCR duplicates were removed. BAM files were used for peak calling with MACS2 callpeak.

## Footprint analysis

Footprint analysis was performed on ATAC-seq data using the TOBIAS suite (v0.17.0). BAM files were first corrected for Tn5 insertion bias using the ATACorrect module with default parameters and the *Homo sapiens* hg19 reference genome. The corrected bigwig files were then analyzed using FootprintScores to compute genome-wide footprint scores. Motif binding prediction and differential binding analysis were conducted with BINDetect, incorporating JASPAR2024 CORE verte-brate transcription factor motifs.

## Differential peak analysis

Two replicates of peak files were combined with IDR and ENCODE blacklist regions were removed. Peaks were annotated by Homer annotatePeaks.pl. The ChIPpeakAnno package in R was used to determine peaks overlapping in two groups. The Diffbind package in R was used to analyze differential binding peaks between two groups. DeepTools was used to create the peak heatmap or profile plot. Homer

v4.11 was used to analyze enrichment of transcription factor binding motifs.

## Active promoter, active enhancer, and super-enhancer identification

We defined H3K27ac-positive regions located within 2 kb upstream or downstream of the transcriptional start site (TSS) as active promoters, while regions positive for both H3K27ac and H3K4me1, but located outside the 2 kb range from the TSS, were classified as active enhancers. Super-enhancers were identified with ROSE package using the enhancer regions.

## HiChIP data analysis

Raw HiChIP sequencing data for H3K27ac in Rv1 cells were downloaded from GEO database (GSE200168). Paired-end HiChIP reads were aligned to the *Homo sapiens* hg19 reference genome using BWA. The pairtools pipeline was used to generate valid pairs and bam files. Contact matrices were constructed using Juicer tools, and chromatin interaction loops were identified with FitHiChIP. WashU Epigenome Browser was used to visualize the loops.

## Bioinformatic analysis

Gene ontology (GO) enrichment analysis (Enrichr web server) was performed using the differentially expressed genes from RNA-seq or the genes associated with annotated ChIP-seq peaks. The Enhanced-Volcano R package was used to create volcano plots. BART web server was used to predict transcriptional regulators associated with the differentially expressed genes. Gene Set Enrichment Analysis (GSEA) was used to determine the enrichment of hallmark pathways among the differentially expressed genes. GEPIA2 web server was used to analyze the correlation of *BPTF* mRNA and AR target mRNAs and patient survival in TCGA PCa dataset. PCTA web server was used to analyze the association between *BPTF* expression and patient survival.

## AR ChIP-seq analysis in patient samples

Normalized AR ChIP-seq bigwig files from primary PCa (n = 18) and CRPC-PDX (n = 15) specimens were obtained from GEO accession GSE130408. Using deepTools (v3.5.6), averaged bigwig files and heatmaps were generated for both primary PCa and CRPC-PDX samples. Signal intensities were calculated from the corresponding bed-graph files. Principal component analysis (PCA) was also performed using deepTools. For heatmap generation, hierarchical clustering was applied, and visualizations were produced using the pheatmap pack-age (v1.0.12). Normalized $\log_2$ values of peak intensities were com-puted for quantification and statistical analysis.

## Prediction of the 3D interaction model by Alphafold

The amino acid sequence of BPTF-BRD and AR-LBD (Fig. S7A) were used as input data on AlphaFold2 with MMseqs2. All input parameters were set to their default values. The top-ranked model predicted by AlphaFold2 was chosen for visualization using ChimeraX (v1.7.1).

*Docking of AU1 in BPTF-BRD*. The structures of BPTF-BRD (pdbID: 7DMY) and AU1 (PubChem CID: 44525934) were downloaded and prepared for docking using AutoDockTools (v1.5.6). AU1 was docked into the BPTF-BRD pocket using AutoDock Vina (v1.1.2) with an energy range of 4 and exhaustiveness of 8. The binding pose of AU1 with the lowest binding energy was chosen and visualized using ChimeraX (v1.7.1). The docked AU1-BRD structure was then superimposed onto the BPTF-BRD/AR-LBD interacting structure (predicted by AlphaFold2) to visualize AU1's potential inhibition of the complex.

## Statistical analysis

RNA-seq was performed in biological triplicates, while ChIP-seq and ATAC-seq were performed in biological duplicates. RIME was per-formed with a single replicate. Other in vitro experiments were

performed in biological triplicate each time and independently repeated at least three times. Biological triplicate from one representative experiment was used for quantification and statistical analysis. Data are presented as means ± SD. Statistical analysis was performed using GraphPad Prism software. Student's t-test (2-tailed) was used to compare differences between two groups of datasets. One-way ANOVA (two-sided) was used to compare the means of more than two groups of datasets. Two-way ANOVA (two-sided) was used when comparing group means involving two independent categorical variables. All multiple comparisons were adjusted using Tukey's test. $P$ values < 0.05 were considered statistically significant.

### Reporting summary

Further information on research design is available in the Nature Portfolio Reporting Summary linked to this article.

## Data availability

The RNA-seq, CUT&RUN ChIP-seq, and ATAC-seq data generated in this study have been deposited in the GEO database under accession code GSE287918. All data are publicly available with no restrictions. The raw sequencing data are available in GEO, and processed data are also included. In addition, publicly available datasets analyzed in this study are as follows: Raw HiChIP sequencing data for H3K27ac in Rv1 cells (GSE200168 [https://www.ncbi.nlm.nih.gov/geo/query/acc.cgi?acc=GSE200168]) and normalized AR ChIP-seq bigwig files from primary PCa and CRPC-PDX specimens (GSE130408 [https://www.ncbi.nlm.nih.gov/geo/query/acc.cgi?acc=GSE130408]). These datasets were reanalyzed to generate figures and results. The mass spectrometry data of AR RIME have been deposited to the ProteomeXchange Consortium via the PRIDE [1] partner repository with the dataset identifier PXD070903. Source data are provided with this paper.

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

## Acknowledgements

This work was supported by Department of Defense grant W81XWH2210363 and National Cancer Institute grant R01CA244667 (to J.Q.). This research was partly supported by funds through the Maryland Department of Health's Cigarette Restitution Fund Program (CH-649-CRF) and the National Cancer Institute—Cancer Center Support Grant—P30CA134274. Part of A.H.'s time was supported by a Merit Review Award (I01 BX000545), Medical Research Service, U.S. Department of Veterans Affairs, and a Baltimore Veterans Affairs Medical Center VALOR Precision Oncology Pilot Award (ID #21PILO02), Prostate Cancer Foundation. The characterization and maintenance of the LuCaP PDX models was supported by the Pacific Northwest Prostate Cancer SPORE grant P50CA97186, NIH P01 grant P01CA163227 and the Institute of Prostate Cancer Research (to E.C.). Part of H.Z.O.'s time was supported by Terry Fox New Frontiers Program Project Grant for supporting Vancouver Prostate Centre members.

## Author contributions

H.Y.J. and J.Q. designed the study. H.Y.J., S.K., M.P., H.R., and J.Q. performed experiments and analyzed the results. M.G. and H.Z.O. provided the prostate cancer tissue microarrays and quantified BPTF immunohistochemistry staining. H.C. provided consultation on statistical analyses. H.M.L. and E.C. provided guidance on accessing and analyzing the RNA-seq datasets from CRPC-PDX specimens. H.Y.J. and J.Q. wrote the manuscript. A.H., X.C., and C.B. provided comments and edits on the manuscript.

## Competing interests

The authors declare no competing interests.
