## [Transparent Peer Review file · Nature Communications]

BPTF regulates androgen receptor activity by enhancing chromatin accessibility and stabilizing the AR-FOXA1 interaction

Corresponding Author: Dr Jianfei Qi

Version 0:

Reviewer comments:

Reviewer #1

(Remarks to the Author)

Jeon et al. study the role of BPTF in prostate cancer. The role of BPTF in other tumor types has been explored but this is a novel contribution. They provide strong evidence of the involvement of BPTF in regulating AR signaling in prostate cancer focusing on the chromatin level. They show rather convincingly that BPTF regulates chromatin accessibility at androgen receptor (AR)-binding sites and promotes AR-FOXA1 complex stabilization. These results open the window to new therapeutic approaches for this tumor given that there is interest in the pharmacological inhibition of BPTF. However, there are significant limitations to the work as reported. Most relevant are:

1. The report of the genomic findings is very skewed towards demonstrating effects on androgen receptor biology (understandable) and dismiss additional biological knowledge that should be derived from a less targeted analysis of the data. It is obvious that a role of BPTF in relationship with AR biology could be highlighted in prostate cancer cells (and not in breast cancer cells, for example). Does this mean that BPTF may "simply" act as an amplifier of lineage identity programs active in the cells? Conceptually, this is a very important point that needs to be clarified and should be discussed. The breadth of the genomics analyses performed is relevant to the field (in fact, the only other dataset of this magnitude available is one preprint dealing with neuroblastoma (PMID 38405949) that the authors should cite, they should also cite another relevant paper on BPTF in leukemia (PMID 37987160). An additional point re: genomic analyses is that statistical significance of the findings is often lacking and this should be fixed.

2. The clinical relevance of the findings is only superficially touched upon and there is a lot of data in the public domain that could be leveraged to address the relevance of BPTF in this tumor, in relationship to androgen resistance.

A general less important point is that it would be useful to mention in the Results section some experimental details (e.g., how many h/days after the KD were the experiments performed). This will allow readers to best assess the relevance of the findings.

Major points

1. BPTF is overexpressed in different tumors and usually associated with bad prognosis. How are BPTF levels in prostate cancer compared with other tumor types? Please, provide more information on the clinical relevance of BPTF.

2. Regarding the cell lines used, how do their levels relate to other published work? A more detailed description of differences between the cell lines used would be important. Also, please justify why most of the experiments are performed with Rv1 cells.

3. The description of the RNA-seq data results should be improved. Please provide information on the most differentially expressed genes and the enriched pathways (e.g., no gene names are provided in volcano plot in Fig. 2B). While the data for AR-dependent genes is convincing, this should be placed in the context of all dysregulated genes. Pathway analysis should be performed. This is particularly relevant because the authors claim that most of the effects observed are independent on MYC (unlike previously reported in other systems), yet several cell cycle-related pathways appear in some of the genomic analyses shown.

4. Expression levels of the AR targets in the cultured cells used should be provided to better evaluate the effect of the BPTF KD.

5. One weakness of the study is the translation of the in vitro data to human disease. Is there any correlation between AR and BPTF in human data? (mRNA and/or protein). Are there differences in BPTF levels according to Gleason score? Is the positive correlation between BPTF expression and the AR signature different or higher in more aggressive stages of

prostate cancer? Could BPTF be considered as an independent factor for prostate cancer survival?

6. Given the established role of MYC in prostate cancer progression (e.g., neuroendocrine switch), the possible implication of MYC in BPTF function should be evaluated in more detail. Although authors stated that "BPTF does not significantly regulate expression or activity of c-MYC in pancreatic cancer cells", it has been proposed that MYC regulates BPTF, and viceversa, in other tumors. Is it possible that MYC does not emerge in their analyses because of biases introduced by the cell lines used in their study?

7. As previously said for RNA-seq, in Cut & Run and ATAC-seq sections, a more comprehensive analysis should be reported. There is no information on the genes annotated to the different regions. Only a selection of motifs and genes in heatmaps are displayed. It would be interesting to know how extensive the overlap of BPTF, AR, and SMARCA1 is at the gene level, the motifs involved, pathways, etc. Does the biology of the BPTF-AR common peaks differ from that of the non-common ones? BPTF has been previously associated to active transcription and promoter regions, how do you explain the low % of promoter signal compared to SE?

8. Why has SMARCA1 KD been chosen for the Cut&Run experiments? It has been described that different NURF complex conformation can be found. It would be interesting to know if BPTF is in the same complex than SMARCA1 or SMARCA5 in PCa cells for downstream analysis. Have the authors performed IP-MS for AR or BPTF to address how significant their physical interaction is in a broader context?

Other points

1. A comment about the specificity of the antibodies used is required.
2. Figure 1J needs to be fixed.
3. Loading controls for the experiments of IP-western (beyond the input) would be useful because in some cases there are differences in the total expression of the proteins analyzed (e.g., FOXA1 in Fig. 6D).
4. It is hard to believe that "there were no significant peaks in the control IgG ChIP... please provide details of thresholds used for peak calling.
5. A title for every figure legend should be included to better deliver the message.
6. While it is true that western blotting for BPTF is not trivial, the quality of the western blots for some of the other proteins could be improved.
7. Immunohistochemical images could be improved.
8. There are a few typos or errors in word usage (e.g., use of "get" instead of "gene" in page 5).
9. Make sure that gene names are written in italics.
10. Include the size (mw) for proteins detected by WB.
11. When describing the genomic distribution for each protein, it would help to include the numbers (%) in the corresponding text section.
12. A supplementary table including the working dilutions for the antibodies used in the study would be useful for the community.
13. AU1 has been reported to inhibit other targets in addition to BPTF and a word of caution about the specificity of this small molecule should be added.
14. A graphical model of the main findings would be useful.

Reviewer #2

(Remarks to the Author)

The NURF complex has a well-established function in remodeling chromatin and modulating the activity of other transcription factors, so it would not be surprising if it made some contribution to AR. Indeed, the investigators should note that a previous study showed that BAP18, which forms an alternative NURF complex with BPTF and SMARCA5 (PMID: 37987160), coactivates AR and promotes PCa (PMID: 27226492). Nonetheless, this study indicates that AR has a particular dependence on NURF (or specifically BPTF), with direct interactions between BPTF, AR, and FOXA1, which is a novel and significant finding. There are a number of points that should be addressed for clarification and to further strengthen the study.

1. Fig 1E should show as well the protein levels of BPTF after knockdown. Fig. 2A and Fig. S2 should similarly show effects on BPTF protein.
2. The RNA-seq analysis in Fig. 2 focuses on genes that are decreased in the BPTF knockdown cells. It would be of interest to similarly assess genes that are increased and whether these are associated with any particularly TFs or pathways. It should also be clarified that the control for the BPTF shRNA is a nontargeting shRNA, and that Fig. 2G reflects biological replicates.
3. The lack of effect on MYC is surprising and may be interesting if it is some special feature of PCa cells. Notably, MYC in PCa cells is regulated by a novel enhancer. It would be of interest to add a positive control in other non-PCa cell lines showing that BPTF does effect MYC.
4. For Fig. 3A, the BPTF input should be included so one can assess the proportion of AR versus BPTF that is precipitated, which would reflect how tightly they are associated. Moreover, since the AR LBD coactivator binding site appears to be the site of interaction, it would be helpful to show effects of androgen and AR antagonists on the interaction.
5. For Fig. 3B, a cartoon showing BPTF structure and location of the fragments would be helpful.
6. In Fig. 4, it would be informative do a similar analysis of the 1,213 AR peaks that are increased after BPTF knockdown.
7. In Fig. 5, it would be of interest to say something about the ATAC peaks that are increased by BPTF knockdown.
8. In Fig. 5C, it appears that the decrease in global ATAC peaks is more than "slight". Moreover, comparing the effects with those at BPTF dependent AR peaks is biased as these are a selected subset that are decreased. Therefore, the appropriate comparison, instead of effects on global ATAC peaks, would be the magnitude of effect on peaks that are decreased. Overall, while the investigators are trying to show BPTF is particularly critical for AR, they need to be more objective in the presentation of the data.

9. While there is some overlap between effects of SMARCA1 and SMARCA5 knockdown, effects on ~1/3 of the genes are in the opposite direction. Moreover, if they were all in the same direction, it is not clear this would suggest redundancy as it would then be required to knockout both to see a robust effect. As noted above, SMARCA5 may be part of a distinct NURF complex that includes BAP18 and BPTF, and this complex has been shown to modulate AR activity. Therefore, this should be noted, and it may be helpful to interpret the data in this context. It would also clearly be of interest to further compare the functions of these NURF complexes, but that is probably beyond the scope of this study.
10. The co-ip in Fig. 6B should also show BPTF input to assess what fraction is being pulled down compared to the fraction of FOXA1.
11. In Fig. 7E, it is unclear what concentration of AU1 is needed to disrupt the AR-BPTF interaction. It would be important to show that this concentration is in the range of the IC50 for effects on AR activity and cell recovery.
12. As the BRD domain appears to mediate the interaction, it may be of interest to assess whether AR.
13. In Fig. 1C and D, exactly what the ADT and NHT samples are should be clarified.
14. In Fig. 1J, one of the bars is misplaced.
15. The figure legends should clarify the statistical tests used.

Reviewer #3

(Remarks to the Author)

Jeon et al. presented the study titled "BPTF regulates androgen receptor activity by enhancing chromatin accessibility and stabilizing the AR-FOXA1 interaction"

Through gene perturbation approaches that inhibit or activate BPTF (shRNA and CRISPRa), they implicate a role of BPTF in prostate cancer cell lines. By examining differentially expressed genes upon BPTF depletion, AR was nominated as an enriched pathway. Through co-IPs, the BPTF protein C-terminal domain interacted with the ligand-binding domain of AR. Through CUT&RUN, they profiled the epigenomic interactions of BPTF, which supported co-regulation of BPTF and AR of promoters and enhancers of AR target genes. BPTF depletion reduced AR interactions with AR regulated sites (ChIP-seq and ATAC-seq). They further indicate that BPTF binding sites are enriched of FOXA1 motifs and may impact SMARCA1 activity. Finally, they conclude that BPTF upregulates AR pathway via FOXA1/SMARCA1.

Overall, the manuscript is beautifully constructed and each hypothesis is evaluated through orthogonal approaches. Minor suggestions would enhance the study.

1. Work has been done on BAP18, a BPTF associated protein. This also indicated a similar role in promoting prostate cancer progression and activates AR. (PMID: 27226492). It also interacts with SMARCA1 and recruits CTCF to ER α -related enhancers in breast cancer (PMID: 36828916). This should be discussed as it related to the central findings.
2. "We measured c-Myc mRNA and protein levels in BPTF-KD PCa cells and found that BPTF KD did not decrease c-Myc mRNA or protein levels in any of the PCa cell lines tested (Figure 2K, S2B)." c-Myc does seem to be upregulated (per WB and mRNA) upon BPTF-KD. Given the oncogenic role of c-Myc towards prostate cancer progression, the authors may want to discuss this finding or exclude this if it is not central to the narrative.
3. CUT&RUN needs to be all uppercase (tradename).
4. The quality photos in the figures of cells and tissue are poor – Fig. 1C, 1F, 1K
5. Fig. 1J – bar plot for Pol II Ab is not aligned.
6. Post shBPTF, the directionality of Fig. 2C-2F is the opposite of Fig. 2G and somewhat counter intuitive. If genes or the AR pathway are suppressed, perhaps the Figures should be revised to reflect these consequences.
7. It would guide the readers to indicate the PHD finger and BRD regions of BPTF in supplement Fig. 3B.
8. Fig 5J – is ZBT10 a control gene? If so, please indicate this through citations and in the Figure legend.
9. Fig. 7I, 7J. The authors indicate "The AR antagonist Enzalutamide (ENZ, 5 μ M) had minimal effect on colony formation". Is this an expected result given some of these cell lines (VCaP, LNCaP) have reported enzalutamide sensitivity?

Reviewer #4

(Remarks to the Author)

Reviewer #5

(Remarks to the Author)

Version 1:

Reviewer comments:

Reviewer #1

(Remarks to the Author)

The authors have adequately addressed most of the issues I had; while I still believe that the analysis is excessively androgen receptor-focused, they have clearly shown that the effects observed are largely independent of the cooperation between BPTF and MYC proteins. Therefore, I am supportive of publication of the current version of the manuscript.

Reviewer #2

(Remarks to the Author)

The authors have responded fully to the points raised in the previous review.

Reviewer #3

(Remarks to the Author)

The authors thorough responses are generally appreciated. My prior concerns have been largely addressed, except the following minor comment:

"Fig.7I, 7J. The authors indicate "The AR antagonist Enzalutamide (ENZ, 5 μ M) had minimal effect on colony formation". Is this an expected result given some of these cell lines (VCaP, LNCaP) have reported enzalutamide sensitivity?"

The authors have responded:

"The reviewer is correct that some studies have reported LNCaP and VCaP cells to be sensitive to enzalutamide 21,22. However, other studies have documented high IC50 values for enzalutamide in these cell lines: For LNCaP cells, reported IC50 values range from 14 μ M to 35 μ M 23-26. Similarly, for VCaP cells, IC50 values range from 9 μ M to as high as 77 μ M 27,28. Consistent with our findings, one study showed that 7.5 μ M enzalutamide had minimal effect on colony formation in LNCaP cells 29. Given the substantial variability in reported enzalutamide sensitivity across studies, our results fall within the expected range and are consistent with some previous studies."

In the revised manuscript:

"To assess the effect of AU1 on PCa cell growth, we performed a colony formation assay using escalating doses of AU1, administered every three days, to determine the IC50. The IC50 values for various PCa cell lines ranged from 2.3 to 3.6 M (Fig. 8G, H, S7D). Enzalutamide (ENZ, 5 M) had minimal effect on colony formation, whereas AU1 (IC50 concentration) inhibited colony formation by ~ 50% (Fig. 8I, J)"

My comment for the revised manuscript:

If the authors are aware of that sub IC50 values are used, then they must explain the reasoning of doing so in the revised manuscript. As stands, it is unclear why AU1 is used at IC50 values whereas enza is not. Please clarify this to the readers.

Reviewer #4

(Remarks to the Author)

Reviewer #6

(Remarks to the Author)

REVIEWER COMMENTS

We sincerely thank the reviewers for their insightful comments and constructive suggestions. In response, we have conducted extensive experiments and made substantial revisions to address the reviewers' critiques. Below is our point-by-point response. Major changes in the manuscript are highlighted in blue.

Reviewer #1 (Remarks to the Author):

Jeon et al. study the role of BPTF in prostate cancer. The role of BPTF in other tumor types has been explored but this is a novel contribution. They provide strong evidence of the involvement of BPTF in regulating AR signaling in prostate cancer focusing on the chromatin level. They show rather convincingly that BPTF regulates chromatin accessibility at androgen receptor (AR)-binding sites and promotes AR-FOXA1 complex stabilization. These results open the window to new therapeutic approaches for this tumor given that there is interest in the pharmacological inhibition of BPTF. However, there are significant limitations to the work as reported. Most relevant are:

1. The report of the genomic findings is very skewed towards demonstrating effects on androgen receptor biology (understandable) and dismiss additional biological knowledge that should be derived from a less targeted analysis of the data. It is obvious that a role of BPTF in relationship with AR biology could be highlighted in prostate cancer cells (and not in breast cancer cells, for example). Does this mean that BPTF may "simply" act as an amplifier of lineage identity programs active in the cells? Conceptually, this is a very important point that needs to be clarified and should be discussed.

Thank you for this important and insightful comment. Our bioinformatic analyses demonstrate that BPTF knockdown (KD) most significantly reduces the AR target gene signature, which ranks #1 across several algorithms (Fig. 2C-2F). While this suggests a primary role in modulating AR signaling, these analyses also indicate that BPTF may influence the activity of additional transcription factors (Fig. 2C) and biological processes (Fig. 2D, 2F).

We have added the following statement in the *Discussion* section: "We demonstrate that BPTF upregulates the expression of a subset of AR hallmark genes through SMARCA1 and/or FOXA1 (Fig. S8B). However, some BPTF-dependent AR target genes are independent of either SMARCA1 or FOXA1 (Fig. S8B), suggesting that BPTF may employ additional mechanisms to regulate AR activity. Our RNA-seq, ChIP-seq and ATAC-seq analyses suggest that BPTF may regulate additional transcription factors or signaling pathways beyond AR signaling. Future works will be necessary to determine whether BPTF regulates additional transcription factors, contributing to both AR-dependent and AR-independent transcriptional programs that drive PCa progression."

Our findings, along with previously published studies, support the idea that BPTF promotes lineage-specific transcriptional programs through distinct mechanisms, rather than simply acting as a general amplifier of existing lineage identity programs. For example, BAP18 (a BPTF-interacting protein) has been shown to facilitate the interaction of BPTF/SMARCA1 with CTCF and ER α in MCF7 breast cancer cells, thereby promoting enhancer-promoter looping and ER α target gene expression ¹. BAP18 can also interact with the BPTF-SMARCA5 complex to regulate the accessibility of insulator regions, thereby promoting CTCF binding and boundary formation, which are critical for maintaining leukemic transcriptional programs ². In our study, we show that BPTF interacts with both AR and FOXA1, two key lineage-specific transcription factors in PCa. However, BPTF significantly enhances AR chromatin binding while having minimal effect on FOXA1 binding at the shared genomic loci (Fig. 4B, 6J). These observations further suggest that BPTF

does not uniformly enhance the binding of lineage-specific transcription factors but instead employs distinct mechanisms to regulate their activity. These points have now been incorporated into the *Discussion* section.

The breadth of the genomics analyses performed is relevant to the field (in fact, the only other dataset of this magnitude available is one preprint dealing with neuroblastoma (PMID 38405949) that the authors should cite, they should also cite another relevant paper on BPTF in leukemia (PMID 37987160).

Thank you for the suggestion. We have now cited both studies to highlight the broader relevance and mechanistic insights of BPTF in other cancer types.

An additional point re: genomic analyses is that statistical significance of the findings is often lacking and this should be fixed.

To address this important point, we have performed statistical analyses and now provided p-values for the following genomic analyses, including Fig. 4A, 4J, 5A, 5C, 5D, 5G, 5H, 7E, 7H, S4A, S4F, S6F, S6M, and S6N.

2. The clinical relevance of the findings is only superficially touched upon and there is a lot of data in the public domain that could be leveraged to address the relevance of BPTF in this tumor, in relationship to androgen resistance.

To address this important critique and enhance the clinical relevance of our findings, we performed additional analyses using publicly available prostate cancer datasets. A summary of both the new and previously presented data is provided below:

- (1) *BPTF* mRNA expression is elevated in castration-resistant prostate cancer (CRPC) compared to primary prostate cancer (PCa) across three independent datasets (Fig. 1A, 1B, S1C).
- (2) BPTF protein levels are higher in CRPC compared to primary PCa tissues (Fig. 1D). Among primary PCa samples, BPTF protein levels are higher in high-grade tumors than low-grade tumors, as defined by Gleason score (Fig. 1E).
- (3) *BPTF* mRNA levels positively correlate with AR target gene signature in TCGA PCa dataset (Fig. 2L).
- (4) High *BPTF* expression tends to associate with shorter patient survival in a prostate cancer dataset (Fig. 7A), although this does not reach statistical significance ($p = 0.115$), likely due to limited sample size.
- (5) The top BPTF signature genes (defined by the most significantly decreased 10 genes in BPTF-knockdown RNA-seq) are associated with shorter disease-free survival in TCGA PCa dataset (Fig. 7B).
- (6) To assess the clinical relevance of BPTF-dependent AR chromatin binding, we analyzed a publicly available AR ChIP-seq data from 18 primary PCa and 15 CRPC-PDX samples (GSE130408), focusing on the AR ChIP-seq peaks at the top 2,000 BPTF-dependent AR-binding sites identified in our study. Principal component analysis (PCA) revealed that these peaks distinguish CRPC-PDX samples from primary PCa samples as two distinct clusters (Fig. 7C), reflecting differential AR peak signal intensity (Fig. 7D, 7E). Hierarchical clustering further stratified CRPC-PDX samples into two subgroups—CRPC-1 and CRPC-2—based on AR peak signal levels: high in CRPC-2 and moderate in CRPC-1 (Fig. 7F-7H). Using RNA-seq data from the same CRPC-PDX samples provided by Dr. Corey, we found that AR activity, as measured by AR scores derived from a 20-gene AR signature, is significantly higher in CRPC-2 than CRPC-1 (Fig. 7K). Importantly, BPTF expression, but not AR expression, is elevated in CRPC-2 relative to CRPC-1 (Fig. 7I, 7J), suggesting that BPTF may enhance AR chromatin binding and transcriptional activity in CRPC-2. Notably, only a subset of BPTF-dependent AR peaks (e.g., peaks ranged 1800–2000) robustly separated AR ChIP-seq samples into three distinct clusters, indicating that select BPTF-dependent AR peaks may have particular utility in classifying PCa subtypes with differing AR activity. Together, these findings support a clinically relevant role for BPTF in regulating AR activity and suggest that

select BPTF-dependent AR binding events may serve as potential biomarkers to stratify PCa subtypes.

A general less important point is that it would be useful to mention in the Results section some experimental details (e.g., how many h/days after the KD were the experiments performed). This will allow readers to best assess the relevance of the findings.

Thank you for the suggestion. We conducted all the experiments 48 hours after shRNA-mediated knockdown, as target gene suppression was not detectable at 24 hours. Efficient knockdown was first observed at 48 hours. We have now included these experimental details in the *Results* section.

Major points

1. BPTF is overexpressed in different tumors and usually associated with bad prognosis. How are BPTF levels in prostate cancer compared with other tumor types? Please, provide more information on the clinical relevance of BPTF.

To address this important critique, we analyzed *BPTF* mRNA expression across 33 different tumor types using the public GEPIA2 web server. The results showed that *BPTF* expression is relatively high in acute myeloid leukemia (LAML) and low in liver hepatocellular carcinoma (LIHC), with prostate adenocarcinoma (PRAD) exhibiting intermediate expression levels comparable to most other cancer types (Fig. S1B). These data support the notion that BPTF is broadly expressed and function as a tumor promoter across multiple tumor types. Notably, the PRAD samples in GEPIA2 consist exclusively of primary prostate cancer. In contrast, our data demonstrate that *BPTF* expression is elevated in CRPC compared to primary PCa (Fig. 1A–1D, S1C). Additional analyses related to the clinical relevance of BPTF are provided in our response to critique #2 above.

2. Regarding the cell lines used, how do their levels relate to other published work? A more detailed description of differences between the cell lines used would be important. Also, please justify why most of the experiments are performed with Rv1 cells.

To address this important critique, we compared relative *BPTF* expression across various cancer cell lines from DepMap Portal. The average *BPTF* mRNA expression in seven PCa cell lines is comparable to that in most other cancer cell lines and ranks 4th among 28 cancer types (Fig. S1A).

BPTF protein levels are comparable among four commonly used PCa cell lines—Rv1, C4-2, LNCaP, and VCaP (Fig. 1G). All four PCa cell lines used in this study are AR-positive. Among them, Rv1 and C4-2 are androgen-independent, whereas LNCaP and VCaP are androgen-dependent. Given our focus on the role of BPTF in CRPC progression, we primarily utilized the androgen-independent Rv1 and C4-2 cell lines. Due to budgetary constraints, we performed genome-wide analyses in only one PCa cell line. We selected the androgen-independent Rv1 cells for these experiments and validated key findings in additional PCa cell lines using complementary approaches, including qRT-PCR, CHIP-qPCR, co-immunoprecipitation, and colony formation assays.

3. The description of the RNA-seq data results should be improved. Please provide information on the most differentially expressed genes and the enriched pathways (e.g., no gene names are provided in volcano plot in Fig. 2B). While the data for AR-dependent genes is convincing, this should be placed in the context of all dysregulated genes. Pathway analysis should be performed. This is particularly relevant because the authors claim that most of the effects observed are independent on MYC (unlike previously reported in other systems), yet several cell cycle-related pathways appear in some of the genomic analyses shown.

Thank you for the valuable suggestion. To provide more comprehensive information on differentially expressed genes (DEGs), we have included the full list of DEGs from the BPTF-KD RNA-seq analysis (Padj

< 0.05, ranked by log₂ fold change; Table S1). To better contextualize AR-regulated genes among the broader set of dysregulated genes, we changed our data presentation to highlight the top 10 downregulated pathways upon BPTF KD. Gene Ontology (GO) analysis revealed that the AR signaling pathway is the most significantly downregulated among these pathways (Fig. 2D). Similarly, Gene Set Enrichment Analysis (GSEA) showed that the AR pathway is the most significantly enriched in control cells relative to BPTF-KD cells (Fig. 2F). These findings support a primary role for BPTF in regulating the AR pathway, while also suggesting it may influence additional pathways—a possibility that warrants future investigation.

In the same analyses, c-Myc is ranked as the 137th transcription factor in BART analysis (Fig. 2C), and the MYC pathway ranked 10th in GO and 23rd in GSEA (Fig. 2D, 2F). BPTF KD had no effect on the c-Myc mRNA or protein levels in four PCa cell lines (Fig. 2K, S2G). A previous study reported that BPTF promotes expression of c-Myc target genes in SW620 colorectal cancer cells³. Consistent with that report, our data show that BPTF KD in SW620 cells reduces expression of representative c-Myc target genes involved in cell cycle and proliferation (Fig. S2L). In contrast, BPTF KD in four prostate cancer cell lines did not significantly affect expression of these genes (Fig. S2H–S2K). These results suggest that the MYC pathway is not a major target of BPTF in prostate cancer cells, although we cannot rule out the possibility that BPTF may regulate some MYC target genes, as suggested by the modest—but detectable—ranking of MYC pathway enrichment in our analyses.

4. Expression levels of the AR targets in the cultured cells used should be provided to better evaluate the effect of the BPTF KD.

We have now included the expression levels of representative AR target genes in the four BPTF-KD PCa cell lines used in this study (Fig. 2H, 2I, S2E).

5. One weakness of the study is the translation of the in vitro data to human disease. Is there any correlation between AR and BPTF in human data? (mRNA and/or protein).

Thank you for this important point. *BPTF* mRNA and *AR* mRNA are positively correlated in TCGA PCa dataset (Fig. S2F). Further, *BPTF* mRNA levels are positively correlated with AR target gene expression in TCGA PCa dataset (Fig. 2L). These in vivo findings are consistent with our in vitro findings that BPTF promotes AR activity by enhancing its chromatin binding. To support the in vivo relevance of the latter finding, our re-analysis of a published AR ChIP-seq studies reveals that *BPTF* mRNA levels are positively correlated with AR chromatin binding and AR activity in CRPC-PDX samples (Fig. 7H-7K).

Are there differences in BPTF levels according to Gleason score?

A subset of primary PCa samples on our TMA slides was categorized based on Gleason score, with scores 6–7 classified as low-grade and scores 8–10 as high-grade. We observed that the H-score of BPTF staining was significantly higher in high-grade PCa compared to low-grade PCa (Fig. 1E).

Is the positive correlation between BPTF expression and the AR signature different or higher in more aggressive stages of prostate cancer?

We found that *BPTF* expression is higher in CRPC compared to primary PCa (Fig. 1A, 1B). In contrast, AR activity—as reflected by AR score—is lower in CRPC than in primary PCa, as shown in the Figure (left). This result is expected, as AR activity in CRPC tissues is suppressed by androgen deprivation therapy (ADT). Although AR signaling is partially reactivated in CRPC compared to metastatic PCa initially treated with ADT, AR activity in CRPC does not return to the levels observed in primary PCa. Therefore, the elevated *BPTF* expression in CRPC does not correspond to increased AR signature expression when compared to primary PCa. However, within CRPC, we observed a positive correlation

between *BPTF* expression and AR signature expression across various CRPC-PDX samples, which were positively correlated with AR chromatin binding (Fig. 7H-7K).

Could BPTF be considered as an independent factor for prostate cancer survival?

We found that high *BPTF* expression was associated with a trend toward shorter overall survival in a PCa dataset, although this did not reach statistical significance, likely due to the limited sample size ($P = 0.115$; Fig. 7A). In contrast, high expression of the BPTF signature—defined as the top 10 genes most significantly upregulated by BPTF—was associated with shorter disease-free survival in TCGA PCa dataset (Fig. 7B). Together, these findings suggest that BPTF or its top signature genes may have potential to serve as independent prognostic markers for prostate cancer outcomes.

6. Given the established role of MYC in prostate cancer progression (e.g., neuroendocrine switch), the possible implication of MYC in BPTF function should be evaluated in more detail. Although authors stated that “BPTF does not significantly regulate expression or activity of c-MYC in prostate cancer cells”, It has been proposed that MYC regulates BPTF, and vice versa, in other tumors. Is it possible that MYC does not emerge in their analyses because of biases introduced by the cell lines used in their study?

We greatly appreciate the reviewer’s important critique. Our data show that BPTF KD in Rv1, C4-2, LNCaP, and VCaP prostate cancer cells did not affect c-Myc protein/mRNA levels (Fig. 2K, S2G) or the expression of representative MYC target genes (Fig. S2H–S2K). In contrast, these MYC target genes were repressed upon BPTF KD in SW620 colorectal cancer cells (Fig. S2L), consistent with the previous report that BPTF regulates c-Myc activity in this context³. These findings suggest that BPTF exerts context-dependent effects on c-Myc activity across different cancer types.

In our RNA-seq of BPTF-KD Rv1 cells, while the AR pathway ranked first among BPTF-upregulated pathways, the MYC pathway ranked 10th in GO and 23rd in GSEA analyses (Fig. 2D, 2F). Based on these findings, we have lowered the tone and removed the statement, “BPTF does not significantly regulate expression or activity of c-MYC in prostate cancer cells”, and replaced it with, “These findings suggest that the MYC pathway is not a primary target of BPTF in PCa cells, although we cannot exclude the possibility that BPTF may regulate a subset of c-Myc target genes, as indicated by the modest—but detectable—enrichment of MYC-related signatures in our analyses.”

7. As previously said for RNA-seq, in Cut &Run and ATAC-seq sections, a more comprehensive analysis should be reported. There is no information on the genes annotated to the different regions. Only a selection of motifs and genes in heatmaps are displayed.

Thank you for this important comment. We have now conducted a more comprehensive analysis of both the CUT&RUN and ATAC-seq datasets, including expanded motif analysis and pathway analysis of genes annotated across different genomic regions. We have also performed and incorporated footprint analysis for the ATAC-seq data.

For the CUT&RUN ChIP-seq of BPTF and AR, we now present the top 20 enriched motifs identified in BPTF-only peaks, BPTF/AR common peaks and AR-only peaks (Fig. S3E-S3G). We also provide the top 10 pathways linked to the genes annotated to each of these peak categories (Fig. S3H–S3J).

Regarding CUT&RUN ChIP-seq of AR in BPTF-KD cells, our previous manuscript focused only on BPTF-enhanced AR peaks and their associated pathways. In the revised manuscript, we have expanded this analysis to include BPTF-repressed AR peaks and the pathways associated with genes annotated to those peaks (Fig. S4C–S4H).

For the ATAC-seq analysis, we now present the genomic distribution of BPTF-enhanced peaks and BPTF-repressed peaks, demonstrating that both sets are predominantly located in intergenic regions (Fig. S5A). Importantly, we performed footprint analysis, which revealed that AR motifs are enriched in global ATAC-seq peaks in control cells compared to BPTF-KD cells (Fig. S5B). Moreover, AR motifs are further enriched in ATAC-seq peaks at the BPTF-dependent AR-binding sites (Fig. S5C). These findings provide further evidence that BPTF enhances chromatin accessibility at these loci to facilitate AR binding.

It would be interesting to know how extensive the overlap of BPTF, AR, and SMARCA1 is at the gene level, the motifs involved, pathways, etc.

To assess the extent of overlap among BPTF, AR, and SMARCA1, we performed CUT&RUN ChIP-seq using a SMARCA1 antibody in Rv1 cells. Differential peak analysis revealed 6,617 genomic regions co-occupied by AR, BPTF and SMARCA1 (Fig. S5G), accounting for 47.2% of AR/BPTF shared peaks.

GO analysis showed that genes associated with AR/BPTF/SMARCA1 peaks (Fig. S5H) were enriched in pathways similar to those associated with AR/BPTF peaks (Fig. S3I). Notably, the androgen response pathway becomes the top enriched pathway for genes associated with AR/BPTF/SMARCA1 peaks (Fig. S5H), compared to its rank of fifth for genes associated with AR/BPTF peaks (Fig. S3I).

The motifs enriched in AR/BPTF/SMARCA1 peaks were largely similar to those in AR/BPTF peaks, with the exception of the FOXA1 motif (Fig. S5I, S3F). Notably, while the FOXA1 motif was strongly enriched at AR/BPTF peaks (ranked 6th; Fig. S3F), it was not among the top motifs at AR/BPTF/SMARCA1 peaks (ranked 27th; Fig. S5I), suggesting that FOXA1 may play a lesser role in recruitment of the AR/BPTF/SMARCA1 complex to these sites. This possibility is consistent with observation that SMARCA1 and FOXA1 display both overlapping and differential effect on the expression of BPTF-dependent AR target genes (Fig. S8B).

Does the biology of the BPTF-AR common peaks differ from that of the non-common ones?

To address this important critique, we performed motif and GO analyses on the distinct peak categories. First, the BPTF-AR common peaks exhibited a unique enrichment profile of transcription factor binding motifs compared to other peak groups (Fig. S3E–S3G). Notably, the canonical AR motif was not among the top enriched motifs in the BPTF-AR common peaks, whereas it was highly enriched in the AR-only peaks. Interestingly, the BPTF-AR common peaks were enriched for motifs such as ROR γ , SP5, SP2, NFY, ELK1, and KLF1—motifs not found in either the BPTF-only or AR-only peaks.

Second, pathway analysis of genes associated with the BPTF-AR common peaks revealed distinct biological processes, including mTORC1 signaling, G2–M checkpoint regulation, and E2F targets, which differed from those associated with the other peak categories (Fig. S3H–S3J). Together, these findings suggest that the BPTF–AR common peaks are functionally and biologically distinct from non-overlapping BPTF or AR peaks.

BPTF has been previously associated to active transcription and promoter regions, how do you explain the low % of promoter signal compared to SE?

We apologize for the confusion. Fig. 4C shows that BPTF-dependent AR peaks are primarily localized at active enhancers (44.9%) and only a small fraction are found at active promoters (7.9%). These findings are consistent with our observation that BPTF knockdown more strongly reduces AR binding at active enhancers than at active promoters (Fig. 4D, S4A).

To address the reviewer's question on BPTF distribution, we performed similar analysis of total BPTF ChIP-seq peaks and found that 42.7% are localized at active promoters, while 24.4% are found at active

enhancers (Fig. 3H, right panel). This distribution pattern supports the reviewer's comment that BPTF has been previously associated with active transcription and promoter regions.

8. Why has SMARCA1 KD has been chosen for the Cut&Run experiments? It has been described that different NURF complex conformation can be found. It would be interesting to know if BPTF is in the same complex than SMARCA1 or SMARCA5 in PCa cells for downstream analysis. Have the authors performed IP-MS for AR or BPTF to address how significant their physical interaction is in a broader context?

Thank you for raising this important question. Our RNA-seq analysis revealed the enrichment of AR pathway in control cells compared with SMARCA1-KD cells (Fig. S5D), whereas no enrichment of AR pathway is found in control cells relative to SMARCA5-KD cells (Fig. S5E). SMARCA1 KD reduced a greater number of BPTF-dependent AR target genes compared with SMARCA5 KD (Fig. 5E, 5F). Based on these observations, we selected SMARCA1 KD for the ATAC-seq experiment (We did not perform CUT&RUN for SMARCA1 KD.)

To further address the reviewer's critique, we provide Supplementary Table S2, which includes AR RIME (Rapid Immunoprecipitation Mass spectrometry of Endogenous proteins) data from Rv1 cells. Over 50% of the AR-interacting proteins identified in our study overlap with those reported in previous mass spectrometry (MS) analyses of the AR complex in prostate cancer cells ⁴⁻⁶, underscoring the reliability of our approach.

BPTF was identified as an AR interactor in our RIME analysis (Table S2). However, similar to previous RIME studies in Rv1 and LN95 cells ⁴, some well-established AR co-factors and co-regulators were not detected, likely due to the relatively low abundance of precipitated AR bait protein. To validate our results, we examined previously published high-quality MS datasets of the AR complex. BPTF was identified in AR complexes in VCaP cells alongside known AR co-factors and co-regulators ⁷. Similarly, SMARCA5 was detected in AR complexes in LNCaP and VCaP cells ^{5,6}.

Collectively, these data support a role for BPTF as a key AR co-activator in prostate cancer cells. Notably, SMARCA1 or SMARCA5 was not simultaneously identified with BPTF in the AR complex, possibly due to the indirect interaction of these ATPases with AR via BPTF. Further optimization of the AR RIME protocol, or alternative chromatin complex isolation approaches, will be needed to more definitively assess the interaction between AR and the NURF complex in prostate cancer cells.

Other points

1. A comment about the specificity of the antibodies used is required.

The specificity of the antibodies used in this study is demonstrated by the Western blot analyses performed in Rv1 and C4-2 cells as shown in the unsliced whole blots above. Each antibody detects a

protein band at the expected molecular weight. In 5% fixed gel: BPTF (~338 kDa). In 5-15% gradient gel: AR (~110 kDa), and the AR splice variant expressed in Rv1 cells (~80 kDa); FOXA1 and c-Myc were detected at ~49 kDa and ~50 kDa, respectively. Importantly, the intensity of each band is reduced upon knockdown of the corresponding target protein, further confirming the specificity of these antibodies.

2. Figure 1J needs to be fixed.

Original Figure 1J (now Fig. 1L) has been fixed.

3. Loading controls for the experiments of IP-western (beyond the input) would be useful because in some cases there are differences in the total expression of the proteins analyzed (e.g., FOXA1 in Fig. 6D).

Thank you for the suggestion. We have now included loading controls for IP-western blot experiments in Fig. 3A, 3E, 3F, 6B, 6D, 8E, S6C, and S6D.

4. It is hard to believe that "there were no significant peaks in the control IgG ChIP... please provide details of thresholds used for peak calling.

We are sorry for the lack of clarity. Peak calling strategies differ between traditional ChIP-seq and CUT&RUN ChIP-seq. In traditional ChIP-seq, peak calling is typically performed by comparing the ChIP sample to an input control, which accounts for the background signals. IgG controls are generally not used for peak calling in traditional ChIP-seq. However, if an IgG control is included, some peaks identified using input controls may also appear in the IgG sample, reflecting non-specific enrichment.

In contrast, CUT&RUN does not use input controls. Instead, it relies on *E. coli* spike-in for normalization and employs IgG controls for peak calling to distinguish specific from non-specific binding. For example, in our CUT&RUN ChIP-seq for BPTF, peaks are called only when there is little to no signal in the corresponding regions of the IgG control. As a result, regions showing strong BPTF peaks exhibit minimal or no signal in the IgG control sample.

We used different p-value thresholds for peak calling in our ChIP-seq experiments based on the immunoprecipitation efficiency of each antibody: $1e-7$ for BPTF, $1e-15$ for AR, and $1e-8$ for FOXA1. These customized thresholds were selected to account for differences in antibody quality and signal-to-noise ratio, ensuring optimal detection of true binding sites for each factor. We applied stringent p-value cutoffs to minimize false positives and retain biologically meaningful peaks. The selected thresholds reflect a balance between sensitivity and specificity and were also guided by the typical number of peaks reported in the literature for each transcription factor^{2,8,9}. This approach allowed us to generate high-confidence peak sets that are robust and comparable across datasets.

5. A title for every figure legend should be included to better deliver the message.

We have added a title for every figure legend.

6. While it is true that western blotting for BPTF is not trivial, the quality of the western blots for some of the other proteins could be improved.

Thank you for the comment. We have reanalyzed previous samples and repeated experiments to improve the quality of the Western blots. As a result, we now provide new, higher-quality Western blot data for all prostate cancer cell samples.

7. Immunohistochemical images could be improved.

We have now provided higher-quality IHC images to replace the previous ones (Fig. 1C).

8. There are a few typos or errors in word usage (e.g., use of "get" instead of "gene" in page 5).

The typos have been corrected.

9. Make sure that gene names are written in italics.

Thank you. All gene symbols have been revised and consistently italicized in the revised manuscript.

10. Include the size (mw) for proteins detected by WB.

Thank you for the suggestion. We have now indicated the molecular weight (MW) of all proteins, except BPTF, in the western blot figures. Notably, BPTF is a very large protein and runs well above the highest MW marker. Its molecular weight can be seen in the submitted Source Data file for the corresponding western blot.

11. When describing the genomic distribution for each protein, it would help to include the numbers (%) in the corresponding text section.

Thank you for the suggestion. We have now included the percentage values in the relevant text and graph describing the genomic distribution of each protein.

12. A supplementary table including the working dilutions for the antibodies used in the study would be useful for the community.

Thank you for the suggestion. We have added Supplementary Table S3, which lists the working dilutions or amounts of antibodies used in this study.

13. AU1 has been reported to inhibit other targets in addition to BPTF and a word of caution about the specificity of this small molecule should be added.

Thank you for pointing this out. We have added a cautionary note in the manuscript: "However, AU1 has been reported to inhibit additional targets beyond BPTF; therefore, potential off-target effects cannot be entirely ruled out¹⁰".

14. A graphical model of the main findings would be useful.

Thank you for the suggestion. We have included graphical models summarizing the key findings of our study (Fig. S8A).

Reviewer #2 (Remarks to the Author):

The NURF complex has a well-established function in remodeling chromatin and modulating the activity of other transcription factors, so it would not be surprising if it made some contribution to AR. Indeed, the investigators should note that a previous study showed that BAP18, which forms an alternative NURF complex with BPTF and SMARCA5 (PMID: 37987160), coactivates AR and promotes PCa (PMID: 27226492). Nonetheless, this study indicates that AR has a particular dependence on NURF (or specifically BPTF), with direct interactions between BPTF, AR, and FOXA1, which is a novel and significant finding. There are a number of points that should be addressed for clarification and to further strengthen the study.

Thank you for reminding us of these important studies. We have now cited and discussed them in the *Discussion* section: “Our findings that BPTF enhances chromatin accessibility, in part through SMARCA1, to facilitate AR binding are consistent with several published studies on BAP18, a BPTF-associated protein. BAP18 has been shown to recruit the MLL1 methyltransferase complex to AR, thereby enhancing AR activity in PCa cells ¹¹. It also interacts with the BPTF–SMARCA1 complex to facilitate CTCF recruitment to ER α , thereby promoting promoter–enhancer looping and ER α target gene expression in breast cancer cells ¹. BAP18 can also interact with the BPTF-SMARCA5 complex to regulate the accessibility of insulator regions, thereby enhancing CTCF binding and boundary formation, which are critical for maintaining leukemic transcriptional programs ². Notably, CTCF is the top enriched transcription factor motif in our BPTF ChIP-seq peaks (Fig. S3B). These findings raise the intriguing possibility that BPTF and BAP18 may cooperate to recruit CTCF to AR-binding sites and regulate promoter–enhancer looping or topologically associated domains (TADs), providing an additional mechanism for modulating AR activity.”

1. Fig 1E should show as well the protein levels of BPTF after knockdown. Fig. 2A and Fig. S2 should similarly show effects on BPTF protein.

Thank you for the suggestion. We have added western blot images showing BPTF protein levels following knockdown in Fig. 1G (for original Fig. 1E), S1E (for original Fig. S2), and S2A (for Fig. 2A).

2. The RNA-seq analysis in Fig. 2 focuses on genes that are decreased in the BPTF knockdown cells. It would be of interest to similarly assess genes that are increased and whether these are associated with any particularly TFs or pathways. It should also be clarified that the control for the BPTF shRNA is a nontargeting shRNA, and that Fig. 2G reflects biological replicates.

Thank you for the great suggestion. To address this, we performed similar analyses on the BPTF-repressed genes (196 upregulated genes following BPTF knockdown, $\log_2[\text{Fold Change}] > 1$, $\text{padj} < 0.05$). BART analysis identified EZH2 as a top-ranked transcriptional regulator potentially regulating these genes (Fig. S2B). Pathway enrichment analysis revealed TNF- α signaling via NF- κ B as the most enriched pathway in GO analysis (Fig. S2C), and AP1 pathway as the top pathway in GSEA analysis (Fig. S2D). These results suggest that BPTF activates or represses distinct transcriptional programs.

We are sorry for the lack of clarification on the shRNA control. We have clarified in the *Methods* section that the control used was a non-targeting shRNA (sequence: TCTCGCTTGGGCGAGAGTAAG). Additionally, we specified in the *Results* section that comparisons were made to non-targeting shRNA controls.

We have also clearly labeled Fig. 2G to show the 3 biological replicates.

3. The lack of effect on MYC is surprising and may be interesting if it is some special feature of PCa cells. Notably, MYC in PCa cells is regulated by a novel enhancer. It would be of interest to add a positive control in other non-PCa cell lines showing that BPTF does effect MYC.

Thank you for this great suggestion. We have included SW620 colorectal cancer cells as a positive control, as BPTF was reported to regulate c-Myc activity in this context ³. Consistently, BPTF knockdown in SW620 cells led to reduced expression of several representative c-Myc target genes (Fig. S2L). In contrast, BPTF knockdown in four prostate cancer cell lines had no apparent effect on the expression of these same c-Myc target genes (Fig. S2H–S2K). These findings support a context-dependent role for BPTF in regulating c-Myc activity.

4. For Fig. 3A, the BPTF input should be included so one can assess the proportion of AR versus BPTF that is precipitated, which would reflect how tightly they are associated. Moreover, since the AR LBD coactivator binding site appears to be the site of interaction, it would be helpful to show effects of androgen and AR antagonists on the interaction.

Thank you for the valuable suggestion. We have included the BPTF input in Fig. 3A. To further examine whether the AR–BPTF interaction is modulated by androgen signaling, we performed co-IP experiments following treatment of C4-2 cells with the synthetic androgen R1881 or the AR antagonist enzalutamide (ENZ). Our results showed that R1881 treatment enhanced the interaction between AR and BPTF, whereas ENZ treatment reduced this interaction (Fig. 3E, 3F), supporting a ligand-dependent regulation of the AR–BPTF complex.

5. For Fig. 3B, a cartoon showing BPTF structure and location of the fragments would be helpful.

Thank you for the suggestion. We have included a cartoon in Fig. S3A illustrating the BPTF protein structure, domain organization, and the locations of the fragments used in this study.

6. In Fig. 4, it would be informative to do a similar analysis of the 1,213 AR peaks that are increased after BPTF knockdown.

Thank you for the helpful suggestion. We have conducted similar analyses of the 1,213 AR peaks that are increased following BPTF knockdown as detailed below.

“Next, we performed similar analyses on the BPTF-repressed AR peaks (i.e., the 1,213 AR peaks enriched in BPTF-KD cells). The heatmap showed increased AR binding following BPTF knockdown (Fig. S4C). Among these peaks, 43.6% were located at active enhancers and 24.5% at active promoters (Fig. S4D), with the increase in AR signal being more pronounced at active enhancers than promoters (Fig. S4E, S4F). BETA analysis revealed a positive correlation between BPTF-KD–enriched AR peaks and genes upregulated after BPTF KD (Fig. S4G). GO analysis of these genes identified skeletal muscle cell differentiation as the top enriched pathway (Fig. S4H). Together, these findings suggest that BPTF directly activates or represses AR binding to distinct subsets of genes, thereby modulating different biological processes.”

7. In Fig. 5, it would be of interest to say something about the ATAC peaks that are increased by BPTF knockdown.

Thank you for the suggestion. We have provided the result and discussion of the ATAC-seq peaks that are increased after BPTF knockdown. “Notably, footprint analysis also revealed increased enrichment of specific transcription factor motifs in BPTF-KD cells compared to control cells (Fig. S5B, S5C, right panels), suggesting that BPTF may also reduce chromatin accessibility to repress the binding of certain transcription factors. Together, these results indicate that BPTF can both enhance and repress chromatin accessibility to regulate the binding of distinct sets of transcription factors. This dual role is consistent with previous reports showing that chromatin remodeling complexes can act as both activator and repressor of gene expression, likely by modulating chromatin structure in either direction—promoting or restricting accessibility¹²⁻¹⁵”

8. In Fig. 5C, it appears that the decrease in global ATAC peaks is more than “slight”. Moreover, comparing the effects with those at BPTF dependent AR peaks is biased as these are a selected subset that are decreased. Therefore, the appropriate comparison, instead of effects on global ATAC peaks, would be the magnitude of effect on peaks that are decreased. Overall, while the investigators are trying to show BPTF is particularly critical for AR, they need to be more objective in the presentation of the data.

Thank you for this important comment. We have revised the description to indicate a “moderate” reduction in global ATAC-seq signal following BPTF KD (Fig. 5C). To objectively evaluate role of BPTF in transcription factor binding in ATAC-seq data, we performed footprint analysis, which detects transcription factor binding motifs within regions protected from Tn5 transposase cleavage due to transcription factor occupancy.

Footprint analysis of total ATAC-seq peaks revealed the enrichment of AR motifs (ranked 10th) in control cells compared with BPTF-KD cells (Fig. S5B, left panel). As expected, footprint analysis of ATAC-seq peaks at BPTF-dependent AR-binding sites showed further enrichment of AR motifs (ranked 4th) in control cells relative to BPTF-KD cells (Fig. S5C, left panel). These results suggest that BPTF promotes AR chromatin binding in control cells by protecting AR motifs from Tn5-mediated cleavage, supporting the hypothesis that BPTF enhances chromatin accessibility to facilitate AR recruitment.

9. While there is some overlap between effects of SMARCA1 and SMARCA5 knockdown, effects on ~1/3 of the genes are in the opposite direction. Moreover, if they were all in the same direction, it is not clear this would suggest redundancy as it would then be required to knockout both to see a robust effect. As noted above, SMARCA5 may be part of a distinct NURF complex that includes BAP18 and BPTF, and this complex has been shown to modulate AR activity. Therefore, this should be noted, and it may be helpful to interpret the data in this context. It would also clearly be of interest to further compare the functions of these NURF complexes, but that is probably beyond the scope of this study.

Thank you for the insightful comment. We agree that SMARCA1 and SMARCA5 regulate approximately two-thirds of the same genes, while SMARCA5 differentially affects the remaining one-third (Fig. 5E, 5F), suggesting that these factors exert both overlapping and distinct effects on the expression of BPTF-dependent AR target genes.

As suggested, to test the potential redundancy between SMARCA1 and SMARCA5 on commonly regulated genes, we analyzed the expression of several AR target genes (*KLK2*, *KLK3*, *STEAP2*) following individual and combined knockdown of SMARCA1 and SMARCA5. Consistent with our RNA-seq data (Fig. 5E, 5F), knockdown of either SMARCA1 or SMARCA5 led to a similar reduction in the expression of these target genes (Fig. S5F). However, the combined knockdown did not result in further suppression beyond that observed with either knockdown alone (Fig. S5F). These findings do not support a redundant role for SMARCA1 and SMARCA5 in regulating these target genes. Instead, the lack of compensatory effects suggests that SMARCA1 and SMARCA5 may operate within distinct NURF complexes to regulate AR activity, as the reviewer astutely noted.

To further address this point, we made extensive efforts to optimize CUT&RUN ChIP-seq for both SMARCA1 and SMARCA5. We successfully performed SMARCA1 ChIP-seq and have incorporated the results into the revised manuscript. However, we were unable to obtain high-quality ChIP-seq data for SMARCA5 due to the lack of suitable antibodies for this application. We appreciate the reviewer's understanding that a comprehensive comparison of the functions of these distinct NURF complexes is beyond the scope of the current study. We plan to pursue this important question in future work.

In response to this critique, we have added the following statement in the *Discussion* section: "SMARCA1 and SMARCA5 are alternative catalytic subunits of the NURF complex¹⁶. Our findings indicate that SMARCA1 and SMARCA5 exert both overlapping and distinct effects on the expression of BPTF-dependent AR target genes, and they do not appear to act redundantly in regulating their shared AR targets. These results suggest that SMARCA1 and SMARCA5 may define distinct NURF complexes—each capable of co-regulating a subset of AR target genes while differentially modulating others. This intriguing possibility warrants further investigation."

10. The co-ip in Fig. 6B should also show BPTF input to assess what fraction is being pulled down compared to the fraction of FOXA1.

We have now included BPTF input in Fig. 6B.

11. In Fig. 7E, it is unclear what concentration of AU1 is needed to disrupt the AR-BPTF interaction. It would

be important to show that this concentration is in the range of the IC₅₀ for effects on AR activity and cell recovery.

We apologize for the lack of clarification. The IC₅₀ of AU1 in the manuscript was determined using a long-term colony formation assay (2 weeks), which yielded values ranging from 2.4 to 3.6 μM in prostate cancer cells (Fig. 8G, 8H). In contrast, for the AR–BPTF interaction assay (Fig. 8E), cells (Rv1 and C4-2) were treated with 5 μM AU1 for 24 hours. It is important to note that IC₅₀ values obtained from short-term viability assays (e.g., 24-hour inhibitor treatment, followed by calculating the percent of survival cells) are typically

Cell line	IC ₅₀ (μM)
Rv1	15.44
C4-2	19.35
VCaP	18.65
LNCaP	11.55

much higher than those derived from long-term colony assays. As shown in the Figure (left), the IC₅₀ of AU1 after 24-hour treatment ranges from ~ 11 to 19 μM across various prostate cancer cell lines. Therefore, the use of 5 μM AU1 for 24 hours to assess AR–BPTF interaction falls well below the IC₅₀ under these conditions and is appropriate for studying its mechanistic effects.

12. As the BRD domain appears to mediate the interaction, it may be of interest to assess whether AR.

The sentence of this critique is truncated. Please help clarify the critique and we will address it.

13. In Fig. 1C and D, exactly what the ADT and NHT samples are should be clarified.

Thank you for pointing this out. NHT (neoadjuvant hormone therapy) is more appropriate than ADT (androgen deprivation therapy) for these samples in Fig. 1C and 1D. We have changed the labeling to NHT.

14. In Fig. 1J, one of the bars is misplaced.

The misplaced bar in original Fig. 1J (now Fig. 1L) has been corrected.

15. The figure legends should clarify the statistical tests used.

The figure legends have now indicated the statistical tests used for each analysis presented in the figure.

Reviewer #3 (Remarks to the Author):

Jeon et al. presented the study titled “BPTF regulates androgen receptor activity by enhancing chromatin accessibility and stabilizing the AR-FOXA1 interaction”

Through gene perturbation approaches that inhibit or activate BPTF (shRNA and CRISPRa), they implicate a role of BPTF in prostate cancer cell lines. By examining differentially expressed genes upon BPTF depletion, AR was nominated as an enriched pathway. Through co-IPs, the BPTF protein C-terminal domain interacted with the ligand-binding domain of AR. Through CUT&RUN, they profiled the epigenomic interactions of BPTF, which supported co-regulation of BPTF and AR of promoters and enhancers of AR target genes. BPTF depletion reduced AR interactions with AR regulated sites (ChIP-seq and ATAC-seq). They further indicate that BPTF binding sites are enriched of FOXA1 motifs and may impact SMARCA1 activity. Finally, they conclude that BPTF upregulates AR pathway via FOXA1/SMARCA1.

Overall, the manuscript is beautifully constructed and each hypothesis is evaluated through orthogonal approaches. Minor suggestions would enhance the study.

1. Work has been done on BAP18, a BPTF associated protein. This also indicated a similar role in promoting prostate cancer progression and activates AR. (PMID: 27226492). It also interacts with SMARCA1 and recruits CTCF to ER α -related enhancers in breast cancer (PMID: 36828916). This should be discussed as it related to the central findings.

Thank you for this insightful suggestion. As recommended, we have added the following discussion in the *Discussion* section:

“Our findings that BPTF enhances chromatin accessibility, in part through SMARCA1, to facilitate AR binding are consistent with several published studies on BAP18, a BPTF-associated protein. BAP18 has been shown to recruit the MLL1 methyltransferase complex to AR, thereby enhancing AR activity in PCa cells ¹¹. It also interacts with the BPTF–SMARCA1 complex to facilitate CTCF recruitment to ER α , thereby promoting promoter–enhancer looping and ER α target gene expression in breast cancer cells ¹. BAP18 can also interact with the BPTF-SMARCA5 complex to regulate the accessibility of insulator regions, thereby enhancing CTCF binding and boundary formation, which are critical for maintaining leukemic transcriptional programs ². Notably, CTCF is the top enriched transcription factor motif in our BPTF ChIP-seq peaks (Fig. S3B). These findings raise the intriguing possibility that BPTF and BAP18 may cooperate to recruit CTCF to AR-binding sites and regulate promoter–enhancer looping or topologically associated domains (TADs), providing an additional mechanism for modulating AR activity.”

2. “We measured c-Myc mRNA and protein levels in BPTF-KD PCa cells and found that BPTF KD did not decrease c-Myc mRNA or protein levels in any of the PCa cell lines tested (Figure 2K, S2B).” c-Myc does seem to be upregulated (per WB and mRNA) upon BPTF-KD. Given the oncogenic role of c-Myc towards prostate cancer progression, the authors may want to discuss this finding or exclude this if it is not central to the narrative.

We apologize for the poor quality of the previous data, as noted by the reviewer. We have now replaced them with high-quality Western blots and qRT-PCR results, which clearly demonstrate that BPTF knockdown has no apparent effect on c-Myc mRNA or protein levels in four prostate cancer cell lines (Fig. 2K, S2G).

3. CUT&RUN needs to be all uppercase (tradename).

Thank you. We have corrected it to uppercase.

4. The quality photos in the figures of cells and tissue are poor – Fig. 1C, 1F, 1K

We have replaced Fig. 1C with the high-resolution tissue images and improved the resolution of cell colony images in Fig. 1H and 1M (original 1F, 1K).

5. Fig. 1J – bar plot for Pol II Ab is not aligned.

We have corrected the issue in original Fig. 1J (now Fig. 1L).

6. Post shBPTF, the directionality of Fig. 2C-2F is the opposite of Fig. 2G and somewhat counter intuitive. If genes or the AR pathway are suppressed, perhaps the Figures should be revised to reflect these consequences.

Thank you for the suggestion. To improve clarity, we revised the *Results* section and title for the graphs to emphasize the use of “BPTF-activated genes” (i.e., genes downregulated following BPTF knockdown) in

the bioinformatic analyses presented in Fig. 2C–2F, which indicate that BPTF most strongly promotes the expression of AR hallmark genes. Consistently, Fig. 5G shows that BPTF knockdown reduces the expression of AR hallmark genes.

7. It would guide the readers to indicate the PHD finger and BRD regions of BPTF in supplement Fig. 3B.

Thank you for the suggestion. We have added a schematic diagram in Fig. S3A to illustrate the structure of the BPTF protein. The diagram also highlights key domains, including the PHD finger and BRD, on BPTF and its truncation mutants.

8. Fig 5J – is ZBT10 a control gene? If so, please indicate this through citations and in the Figure legend.

We apologize for the lack of clarification. ZBTB10 is one of the 20 experimentally validated AR transcriptional target genes¹⁷. The expression patterns of these genes are widely used to calculate the AR score, a metric commonly employed to assess AR transcriptional activity in prostate cancer tissues¹⁸⁻²⁰. In our study, we also used this 20-gene AR target signature to calculate the AR score in the CPRC-PDX tissues (Fig. 7K).

As suggested, we indicate this in the legend of Fig. 5J with citation: “ZBTB10 is an experimentally validated AR target gene¹⁷.”

9. Fig.7I, 7J. The authors indicate “The AR antagonist Enzalutamide (ENZ, 5 μ M) had minimal effect on colony formation”. Is this an expected result given some of these cell lines (VCaP, LNCaP) have reported enzalutamide sensitivity?

The reviewer is correct that some studies have reported LNCaP and VCaP cells to be sensitive to enzalutamide^{21,22}. However, other studies have documented high IC₅₀ values for enzalutamide in these cell lines: For LNCaP cells, reported IC₅₀ values range from 14 μ M to 35 μ M²³⁻²⁶. Similarly, for VCaP cells, IC₅₀ values range from 9 μ M to as high as 77 μ M^{27,28}. Consistent with our findings, one study showed that 7.5 μ M enzalutamide had minimal effect on colony formation in LNCaP cells²⁹. Given the substantial variability in reported enzalutamide sensitivity across studies, our results fall within the expected range and are consistent with some previous studies.

Reviewer #4 (Remarks to the Author):

Reviewer #5 (Remarks to the Author):

REFERENCE

- 1 Sun, G. *et al.* BAP18 facilitates CTCF-mediated chromatin accessible to regulate enhancer activity in breast cancer. *Cell Death Differ* **30**, 1260-1278 (2023). <https://doi.org:10.1038/s41418-023-01135-y>
- 2 Radzisheuskaya, A. *et al.* An alternative NURF complex sustains acute myeloid leukemia by regulating the accessibility of insulator regions. *EMBO J* **42**, e114221 (2023). <https://doi.org:10.15252/embj.2023114221>
- 3 Guo, P. *et al.* BPTF inhibition antagonizes colorectal cancer progression by transcriptionally inactivating Cdc25A. *Redox Biol* **55**, 102418 (2022). <https://doi.org:10.1016/j.redox.2022.102418>
- 4 Liang, J. *et al.* Androgen receptor splice variant 7 functions independently of the full length receptor in prostate cancer cells. *Cancer Lett* **519**, 172-184 (2021). <https://doi.org:10.1016/j.canlet.2021.07.013>
- 5 Yokoyama, A. *et al.* Identification and Functional Characterization of a Novel Androgen Receptor Coregulator, EAP1. *J Endocr Soc* **5**, bvab150 (2021). <https://doi.org:10.1210/jendso/bvab150>
- 6 Chen, M. *et al.* TRIM33 drives prostate tumor growth by stabilizing androgen receptor from Skp2-mediated degradation. *EMBO Rep* **23**, e53468 (2022). <https://doi.org:10.15252/embr.202153468>
- 7 Launonen, K. M. *et al.* Chromatin-directed proteomics-identified network of endogenous androgen receptor in prostate cancer cells. *Oncogene* **40**, 4567-4579 (2021). <https://doi.org:10.1038/s41388-021-01887-2>
- 8 Gao, S. *et al.* Chromatin binding of FOXA1 is promoted by LSD1-mediated demethylation in prostate cancer. *Nat Genet* **52**, 1011-1017 (2020). <https://doi.org:10.1038/s41588-020-0681-7>
- 9 Augello, M. A. *et al.* CHD1 Loss Alters AR Binding at Lineage-Specific Enhancers and Modulates Distinct Transcriptional Programs to Drive Prostate Tumorigenesis. *Cancer Cell* **35**, 817-819 (2019). <https://doi.org:10.1016/j.ccell.2019.04.012>
- 10 Sinanian, M. M. *et al.* A BPTF Inhibitor That Interferes with the Multidrug Resistance Pump to Sensitize Murine Triple-Negative Breast Cancer Cells to Chemotherapy. *Int J Mol Sci* **25** (2024). <https://doi.org:10.3390/ijms252111346>
- 11 Sun, S. *et al.* BAP18 coactivates androgen receptor action and promotes prostate cancer progression. *Nucleic Acids Res* **44**, 8112-8128 (2016). <https://doi.org:10.1093/nar/gkw472>
- 12 Morse, K., Bishop, A. L., Swerdlow, S., Leslie, J. M. & Unal, E. Swi/Snf chromatin remodeling regulates transcriptional interference and gene repression. *Mol Cell* **84**, 3080-3097 e3089 (2024). <https://doi.org:10.1016/j.molcel.2024.06.029>
- 13 Schnitzler, G., Sif, S. & Kingston, R. E. Human SWI/SNF interconverts a nucleosome between its base state and a stable remodeled state. *Cell* **94**, 17-27 (1998). [https://doi.org:10.1016/s0092-8674\(00\)81217-9](https://doi.org:10.1016/s0092-8674(00)81217-9)
- 14 Holstege, F. C. *et al.* Dissecting the regulatory circuitry of a eukaryotic genome. *Cell* **95**, 717-728 (1998). [https://doi.org:10.1016/s0092-8674\(00\)81641-4](https://doi.org:10.1016/s0092-8674(00)81641-4)
- 15 Tyler, J. K. & Kadonaga, J. T. The "dark side" of chromatin remodeling: repressive effects on transcription. *Cell* **99**, 443-446 (1999). [https://doi.org:10.1016/s0092-8674\(00\)81530-5](https://doi.org:10.1016/s0092-8674(00)81530-5)

- 16 Li, Y. *et al.* The emerging role of ISWI chromatin remodeling complexes in cancer. *J Exp Clin Cancer Res* **40**, 346 (2021). <https://doi.org/10.1186/s13046-021-02151-x>
- 17 Hieronymus, H. *et al.* Gene expression signature-based chemical genomic prediction identifies a novel class of HSP90 pathway modulators. *Cancer Cell* **10**, 321-330 (2006). <https://doi.org/10.1016/j.ccr.2006.09.005>
- 18 Hu, C., Fang, D., Xu, H., Wang, Q. & Xia, H. The androgen receptor expression and association with patient's survival in different cancers. *Genomics* **112**, 1926-1940 (2020). <https://doi.org/10.1016/j.ygeno.2019.11.005>
- 19 Stelloo, S. *et al.* Integrative epigenetic taxonomy of primary prostate cancer. *Nat Commun* **9**, 4900 (2018). <https://doi.org/10.1038/s41467-018-07270-2>
- 20 Cancer Genome Atlas Research, N. The Molecular Taxonomy of Primary Prostate Cancer. *Cell* **163**, 1011-1025 (2015). <https://doi.org/10.1016/j.cell.2015.10.025>
- 21 Li, S. *et al.* Activation of MAPK Signaling by CXCR7 Leads to Enzalutamide Resistance in Prostate Cancer. *Cancer Res* **79**, 2580-2592 (2019). <https://doi.org/10.1158/0008-5472.CAN-18-2812>
- 22 Asangani, I. A. *et al.* BET Bromodomain Inhibitors Enhance Efficacy and Disrupt Resistance to AR Antagonists in the Treatment of Prostate Cancer. *Mol Cancer Res* **14**, 324-331 (2016). <https://doi.org/10.1158/1541-7786.MCR-15-0472>
- 23 Hellsten, R., Stiehm, A., Palominos, M., Persson, M. & Bjartell, A. The STAT3 inhibitor GPB730 enhances the sensitivity to enzalutamide in prostate cancer cells. *Transl Oncol* **24**, 101495 (2022). <https://doi.org/10.1016/j.tranon.2022.101495>
- 24 Liadi, Y. M., Campbell, T., Hwang, B. J., Elliott, B. & Odeero-Marah, V. High Mobility Group AT-hook 2: A Biomarker Associated with Resistance to Enzalutamide in Prostate Cancer Cells. *Cancers (Basel)* **16** (2024). <https://doi.org/10.3390/cancers16152631>
- 25 Haldrup, J., Weiss, S., Schmidt, L. & Sorensen, K. D. Investigation of enzalutamide, docetaxel, and cabazitaxel resistance in the castration resistant prostate cancer cell line C4 using genome-wide CRISPR/Cas9 screening. *Sci Rep* **13**, 9043 (2023). <https://doi.org/10.1038/s41598-023-35950-7>
- 26 Wu, L. J. *et al.* Synergistic combination therapy with ONC201 or ONC206, and enzalutamide or darolutamide in preclinical studies of castration-resistant prostate cancer. *Am J Cancer Res* **14**, 6012-6036 (2024). <https://doi.org/10.62347/VJMW4904>
- 27 Goode, E. A. *et al.* Sialylation Inhibition Can Partially Revert Acquired Resistance to Enzalutamide in Prostate Cancer Cells. *Cancers (Basel)* **16** (2024). <https://doi.org/10.3390/cancers16172953>
- 28 Biernacka, K. M., Barker, R., Sewell, A., Bahl, A. & Perks, C. M. A role for androgen receptor variant 7 in sensitivity to therapy: Involvement of IGFBP-2 and FOXA1. *Transl Oncol* **34**, 101698 (2023). <https://doi.org/10.1016/j.tranon.2023.101698>
- 29 Fletcher, K. A., Alkurashi, M. H. & Lindsay, A. J. Endosomal recycling inhibitors downregulate the androgen receptor and synergise with enzalutamide. *Invest New Drugs* **42**, 14-23 (2024). <https://doi.org/10.1007/s10637-023-01407-x>

REVIEWER COMMENTS

Reviewer #3 (Remarks to the Author):

The authors thorough responses are generally appreciated. My prior concerns have been largely addressed, except the following minor comment:

"Fig.7I, 7J. The authors indicate "The AR antagonist Enzalutamide (ENZ, 5 μ M) had minimal effect on colony formation". Is this an expected result given some of these cell lines (VCaP, LNCaP) have reported enzalutamide sensitivity?"

The authors have responded:

"The reviewer is correct that some studies have reported LNCaP and VCaP cells to be sensitive to enzalutamide 21,22. However, other studies have documented high IC₅₀ values for enzalutamide in these cell lines: For LNCaP cells, reported IC₅₀ values range from 14 μ M to 35 μ M 23-26. Similarly, for VCaP cells, IC₅₀ values range from 9 μ M to as high as 77 μ M 27,28. Consistent with our findings, one study showed that 7.5 μ M enzalutamide had minimal effect on colony formation in LNCaP cells 29. Given the substantial variability in reported enzalutamide sensitivity across studies, our results fall within the expected range and are consistent with some previous studies."

In the revised manuscript:

"To assess the effect of AU1 on PCa cell growth, we performed a colony formation assay using escalating doses of AU1, administered every three days, to determine the IC₅₀. The IC₅₀ values for various PCa cell lines ranged from 2.3 to 3.6 μ M (Fig. 8G, H, S7D). Enzalutamide (ENZ, 5 μ M) had minimal effect on colony formation, whereas AU1 (IC₅₀ concentration) inhibited colony formation by ~ 50% (Fig. 8I, J)"

My comment for the revised manuscript:

If the authors are aware of that sub IC₅₀ values are used, then they must explain the reasoning of doing so in the revised manuscript. As stands, it is unclear why AU1 is used at IC₅₀ values whereas enza is not. Please clarify this to the readers.

We sincerely thank the reviewer for raising this important point, which helped us recognize an error in our previous discussion of the enzalutamide results. We apologize for not identifying this issue in our earlier response.

Specifically, the reviewer's previous comments regarding the sensitivity of LNCaP and VCaP cells to enzalutamide are correct. As shown in Figure 8J, 5 μ M enzalutamide inhibited colony formation of androgen-sensitive LNCaP and VCaP cells by ~43% and ~20%, respectively, but had less effect on androgen-insensitive C4-2 cells (~16% inhibition) and no effect on enzalutamide-resistant Rv1 cells. Thus, as expected, LNCaP and VCaP cells are more responsive to enzalutamide, whereas C4-2 and Rv1 cells are less or not responsive.

Notably, 5 μ M enzalutamide represents the IC₅₀ concentration for inhibiting colony formation of LNCaP cells and a sub-IC₅₀ concentration for Rv1, C4-2, and VCaP cells. However, when combined with the IC₅₀ concentration of AU1, enzalutamide (5 μ M) produced a synergistic effect, reducing colony formation of Rv1, C4-2, and VCaP cells by ~90%. These findings demonstrate that AU1 can sensitize these PCa cells to enzalutamide, even at a sub-IC₅₀ concentration that is otherwise ineffective. This effect may be explained by AU1 reducing AR activity in PCa cells, thereby enhancing their sensitivity to enzalutamide.

We have added p-values comparing control and enzalutamide treatment to Figure 8J and revised the text accordingly:

“Enzalutamide (ENZ, 5 μ M) inhibited colony formation of androgen-sensitive LNCaP and VCaP cells by ~43% and ~20%, respectively, but had less effect on androgen-insensitive C4-2 cells (~16% inhibition) and no effect on enzalutamide-resistant Rv1 cells (Fig. 8I, J). In contrast, AU1 (IC₅₀ concentration) inhibited colony formation of all PCa cell lines by ~50% (Fig. 8I, J). Notably, the combination of enzalutamide (5 μ M) and AU1 (IC₅₀ concentration) synergistically inhibited Rv1, C4-2, and VCaP cells, reducing colony formation by ~90% (Fig. 8I, J). These results indicate that AU1 can sensitize these PCa cells to enzalutamide, even at a sub-IC₅₀ concentration that is ineffective on its own.”